# Assessing Women’s Menstruation Concerns and Experiences in Rural India: Development and Validation of a Menstrual Insecurity Measure

**DOI:** 10.3390/ijerph17103468

**Published:** 2020-05-15

**Authors:** Bethany A. Caruso, Gerard Portela, Shauna McManus, Thomas Clasen

**Affiliations:** 1Hubert Department of Global Health, Rollins School of Public Health, Emory University, Atlanta, GA 30322, USA; 2Department of Epidemiology, Rollins School of Public Health, Emory University, Atlanta, GA 30322, USA; gerard.thomas.portela@emory.edu; 3Department of Biostatistics and Bioinformatics, Rollins School of Public Health, Emory University, Atlanta, GA 30322, USA; shauna.mcmanus@emory.edu; 4Gangarosa Department of Environmental Health, Rollins School of Public Health, Emory University, Atlanta, GA 30322, USA; thomas.f.clasen@emory.edu

**Keywords:** gender, security, factor analysis, management, restriction, pain, menstrual health

## Abstract

Qualitative research has documented menstruator’s challenges, particularly in water and sanitation poor environments, but quantitative assessment is limited. We created and validated a culturally-grounded measure of Menstrual Insecurity to assess women’s menstruation-related concerns and negative experiences. With cross-sectional data from 878 menstruating women in rural Odisha, India, we carried out Exploratory (EFA) and Confirmatory (CFA) Factor Analyses to reduce a 40-item pool and identify and confirm the scale factor structure. A 19-item, five factor model best fit the data (EFA: root mean square error of approximation (RMSEA) = 0.027; comparative fit index (CFI) = 0.994; Tucker-Lewis index (TLI) = 0.989; CFA: RMSEA = 0.058; CFI = 0.937; TLI = 0.925). Sub-scales included: Management, Menstrual Cycle Concerns, Symptoms, Restrictions, and Menstruation-Related Bodily Concerns. Those without access to a functional latrine, enclosed bathing space, water source within their compound, or who used reusable cloth had significantly higher overall Menstrual Insecurity scores (greater insecurity) than those with these facilities or using disposable pads. Post-hoc exploratory analysis found that women reporting experiencing tension at menstrual onset or difficulty doing work had significantly higher Menstrual Insecurity scores. This validated tool is useful for measuring Menstrual Insecurity, assessing health inequities and correlates of Menstrual Insecurity, and informing program design.

## 1. Introduction

Increasing efforts to query, document, and share people’s lived experiences of menstruation have resulted in recognition of menstruation as a public health issue [1], galvanized investment in and implementation of programs and policies [2,3], and inspired research priorities [4] and multi-sector collaboration and vision setting to improve circumstances and assessment strategies globally [5,6,7]. An ever-growing body of qualitative research, primarily focused on girls’ menstruation experiences at school in low and middle income countries (LMICs), has shaped current understanding of how menstruation is experienced [8]. While qualitative research importantly provides detail of menstruation experiences, it is undertaken with small samples and cannot reveal how common and frequent experiences are among more representative populations. Quantitative assessment of menstruators’ experiences is needed to elucidate priorities and inform programming and policy. The aim of this paper is to detail the process of creating a ground-up, locally-derived measure of Menstrual Insecurity, a construct that aims to represent the suite of experiences and concerns about and of menstruation, among women in rural Odisha, India.

A 2019 systematic review synthesizing findings from 87 qualitative publications on women’s and girl’s menstruation experiences in LMICs illuminates how these menstruation experiences can have broader impacts [8]. Specifically, their conceptual model illustrates that an array of antecedents (e.g., sociocultural context, resources, knowledge, environments) influence menstruation experiences (e.g., menstrual practices, confidence managing menstruation, shame and distress, symptoms), which in turn can impact social participation, education, employment, and psychological and physical health [8]. The authors highlight that some interventions have aimed to influence health and education (like school attendance) by improving antecedents (like knowledge or product access), without considering how experiences are important mediators of these relationships. The authors emphasize the need to develop measures of menstruation experience. The present research contributes a measure to assess menstruation experiences, which can be used independently and to determine how experiences influence health, employment, or other impacts.

There is limited understanding of women’s experiences of menstruation in LMICs and the specific challenges and concerns they face. While the concerns that adults and adolescents report overlap greatly, adult health needs and experiences may differ through the life course, could result in different impacts on well-being and life outcomes, and are worthy of their own attention [9]. Notably, women in Odisha, India report that managing menstruation is a source of psychosocial stress [10] and that it is the most stressful of their sanitation-related behaviors [11]. Only 18% (16 of 87) of the publications included in the review focused exclusively on women [8]. Women, from studies included both in the review and not, discuss the taboo and secretive nature of menstruation, behavioral expectations, varied restrictions, lack of practical or accurate biological information about menstruation, insufficient facilities and resources for management, and concerns about health due to menstrual patterns, symptoms, and flow [10,11,12,13,14,15,16,17,18,19,20,21,22,23].

Limited quantitative studies have documented experiences, behaviors, or concerns, like needing to abide by behavioral restrictions, having options for changing menstrual materials, or depending on others, like sexual partners, for materials [24,25,26]. However, menstruators do not have one concern or challenge alone. Women participating in qualitative interviews in rural Odisha, where the present research took place, reported 32 unique menstruation concerns that they believed women in their community experienced, including concerns about bathing at menstrual onset (52%), washing (51%) and drying (46%) menstrual cloths, feeling general discomfort (43%), being seen by people while managing menses (37%), experiencing pain (34%), feeling dirty (30%), dealing with behavior restrictions (30%), and worrying about irregularity (27%) [22]. These data demonstrate that women’s concerns include and extend beyond management to needs [12] and support the need to understand how prevalent and frequent an array of experiences and concerns are at the individual level in a broader population.

To fill the need to understand prevalence and frequency of menstruator’s concerns and experiences, we aimed to develop a culturally-grounded measure to assess the suite of concerns and negative menstruation-related experiences that women in rural Odisha voiced. The measure is inclusive of—but not exclusive to—experiences managing menses; a scale assessing menstrual needs among adolescents has been created [27]. This broad approach, inclusive of participant-identified concerns and experiences, will enable understanding of the range of issues women in this population have and can enable future assessments of how concerns and experiences may impact menstruator’s lives and inform program development.

## 2. Materials and Methods

### 2.1. Defining Menstrual Insecurity

We refer to the complement of concerns and negative menstruation-related experiences women in this population face as ‘Menstrual Insecurity’. Following best practice [28], we offer a preliminary definition of Menstrual Insecurity to be ‘the suite of social, environmental, and biological concerns and negative experiences resulting from menstruation’. Our intentionally broad, preliminary conceptualization is informed by efforts to create culturally-grounded measures of food [29,30,31], water [32,33,34], and sanitation [35] insecurity. Researchers in these fields argue for extending understanding of insecurity constructs beyond biological needs to include the broader socio-cultural context, like social needs and experiences, which may impact well-being [31,32,34,35,36]. Our definition is intentionally broad and described as preliminary as we consider Menstrual Insecurity to be a concept to be iterated upon as research evolves, which has been demonstrated with the concepts of Water Insecurity [37] and Menstrual Hygiene Management [12,23].

### 2.2. Setting

Data for this research was collected in rural Puri district, Odisha, India, concurrently with data collected to create a measure of Sanitation Insecurity (March 2014–February 2015) [35]. At the time of data collection, 61% of rural inhabitants in India were estimated to practice open defecation [38]. In the India National Family Health Survey Report (NFHS-4, 2015–2016), which includes data collected just after the data from the present study, 71% of women age 15–24 (the only age range queried) from rural India used cloth for menstruation, 15% used locally prepared napkins, and 37% used sanitary pads (multiple methods of use could be reported) [39]. A case-control study in Odisha found that women who used reusable cloths for menstruation were more likely to report symptoms of urogenital infection or have a lab-confirmed diagnosis of either bacterial vaginosis or a urinary tract infection than those using disposable pads [40].

Research was conducted in villages previously engaged in a cluster randomized controlled trial (May 2010–December 2013) to assess the health impacts of a latrine provision intervention [41,42,43]. While sanitation coverage increased in intervention villages, no reductions in diarrhea, soil-transmitted helminth infection, or child malnutrition attributable to the intervention were identified [41].

### 2.3. Overview of Research Design

The present research employed the mixed-methods approach that was used to create the measure of Sanitation Insecurity [35]. The approach leverages the perspectives and experiences of women, and is based on the ground-up, inductive approaches used to create measures of food and water insecurity [30,36]. The three-phase, sequential, mixed-methods design included a qualitative phase to understand the suite of women’s menstruation concerns and experiences, a quantitative phase to capture the frequency and intensity of those concerns, and a measurement finalization phase involving various statistical analyses (See Figure 1).

This measure was intentionally developed separately from the measure of Sanitation Insecurity, which only included items related to urination and defecation [35]. The Sanitation Insecurity measure was intended to be applicable to all women. Because not all women menstruate, due to pregnancy, recent birth, and life stage, among other reasons, menstruation items were not included. Further, qualitative work identified menstruation concerns and experiences beyond those related to sanitation [12,22]. This Menstrual Insecurity measure can be used independently or in tandem with the Sanitation Insecurity measure.

### 2.4. Phase 1: Qualitative Research

Phase 1 involved qualitative data collection, preliminary measurement item identification, and review and finalization of items for phase 2 survey administration.

#### 2.4.1. Phase 1, Stage 1: Data Collection

Qualitative data collection included Free-listing Interviews (FLIs) and Focus Group Discussions (FGDs). FLIs were held first to inductively identify concerns and experiences women had related to menstruation to inform items for the Menstrual Insecurity measure (March–April 2014). Free-listing is a methodology used among homogenous groups, in this case rural women from Odisha, India, to identify shared perceptions about a topic [44]. We asked women to list their concerns about menstruation and probed them to identify additional concerns at night or during the monsoon season. Interviews were conducted with women from eight villages (5 former intervention, 3 former control). Sixty-nine women were purposively selected for one-on-one interviews from four life stages: (1) unmarried (*n* = 16); (2) married three years or less (*n* = 12); (3) married over three years (*n* = 22); (4) and women over 49 years of age (*n* = 19). These categories were intentionally selected given assumed variation in their sanitation and hygiene experiences [22]. Women no longer menstruating provided insights about when they were menstruating.

We then held eight FGDs (*n* = 46) with women from four different villages (2 former intervention, 2 former control), four with unmarried women (*n* = 23) and four with married women (*n* = 23) (April–May 2014). During the FGDs, we first asked women to share their perceptions of the concerns that ‘women in their community’ had about menstruation to capture new insights and not compel women to share personal information. We then asked for detail about the intensity and frequency of those concerns, and about concerns described in the FLIs if not already mentioned in the FGDs.

FLIs and FGDs were conducted in Oriya with trained research assistants. Each was recorded, then translated and transcribed into English. Further information about the qualitative methods, the participants engaged, and the findings related to sanitation insecurity [22] broadly and menstruation [12] specifically are reported elsewhere.

#### 2.4.2. Phase 1, Stage 2: Item Identification

All FLIs were analyzed first to identify the full suite of concerns or negative experiences participants believed women in their community to have. We then determined the proportion of participants who reported each concern or experience. Thirty-two unique menstruation concerns were shared by 67 (97%) of the 69 women interviewed. FGDs supported information gathered in FLIs (See Caruso 2017, including Appendix A, for list and frequencies of concerns) [22]. Next, concerns were analyzed thematically to identify emergent grouping. We identified and sorted concerns into four themes, which included: Management, Restrictions or challenges to normal activities, Social needs and constraints, and Well-Being. Wording of the identified concerns in each theme was then adapted to create draft items. (See Appendix A for Survey Items by theme.) The initial list of items included the full range of concerns noted, however monsoon-specific items were omitted due to irrelevance during the following anticipated data collection period. We hypothesized that these themes would emerge as factors in the final measure.

#### 2.4.3. Phase 1, Stage 3: Item Review and Finalization

To finalize the pool of items, we conducted four rounds of review. First, to assess content validity, feedback on items was solicited from two external experts with experience research in sanitation and menstruation [28,45]. Second, the two research assistants who conducted the FLIs and FGDs provided additional assessment of content validity, specifically face validity, by confirming that items captured the range of concerns they heard and that wording was appropriate. The two RAs independently translated the items, then compared the translations item-by-item, and discussed alternative phrasings before finalizing translation. Third, as a preliminary means of pre-testing, each item was reviewed with the nine female enumerators hired to administer the survey using a cognitive interview approach [28,46]. In addition to having experience with survey data collection, including studies specific to sanitation and hygiene, all enumerators were from villages similar to those to be engaged in the next phase and were able to comment on the items regarding item content and translation. For each item, the RAs asked a team member to explain what the question was asking in their own words. Other team members were welcome to comment or offer an alternative understanding. The group then discussed, and changes to the item translation were made as needed. Fourth, the RAs and enumerator team piloted the items in a village like those to be surveyed. Team members noted any items that posed challenges for participants and suggested alternative phrasing. Final changes were made during a team meeting after the pilot.

The final tool included 40 items that asked women how often they had a particular menstruation-related experience during their last two menstrual periods: never, sometimes, often, or always (See Appendix B for the tool).

### 2.5. Phase 2: Quantitative Research

Quantitative research involved a census to create sampling frames, the creation of sampling frames and final sampling lists, and survey administration (See Figure 1).

#### 2.5.1. Phase 2, Stage 1: Household Census

To identify eligible participants, we conducted a paper census in 60 of the former 100 trial villages. We aimed to survey 1440 participants from those 60 villages (30 former controls and 30 former recipients of the intervention) via a stratified, multi-stage, cluster sample design. Our sample size was powered for the parent study to detect effect sizes of sanitation insecurity on mental health outcomes (well-being, anxiety, depression, and distress) using a multilevel modeling (hierarchical modeling) approach with consideration for two levels: cluster level (e.g., former intervention status of the village) and individual level (e.g., latrine access, life stage, etc.) [47]. Results for that analysis are reported elsewhere [48]. We determined our base sample size of 60 clusters with 20 participants each (1200 participants) from a simulation study that demonstrated power to detect small (*d* = 0.20) direct and cross-level interaction effects for a continuous level-2 predictor to be greater than 96% [49]. We targeted 1440 participants (24 per village) to allow for attrition from: incomplete data, census errors misidentifying eligible participants, and unintentional multiple sampling of households.

We used the former intervention status of the village as a means of achieving sanitation coverage variability in our sample. Villages that had previously received the intervention were eligible if they had greater than 25% larine coverage. We determined coverage from the final trial data collection effort (December 2014) [41] assuming that coverage would not have changed markedly in the nine months since. Former control villages were eligible for inclusion if latrine coverage was less than 20%. We gained feedback from a local non-government organization in Puri working to provide latrines in the former control villages to discern eligibility. Villages were ineligible if they had been engaged in the qualitative activities informing survey development.

The trained enumerator team visited each household in each eligible village to administer the census. The enumerator asked a representative from each of household to share information about household members, like sex, age, and marital status, and household water and sanitation access.

#### 2.5.2. Phase 2, Stage 2: Creation of Sampling Frames and Final Sampling Lists

We created sampling frames from village census data to enable random selection of participants by life stage category. We sought women over age 18 from the four life stages previously noted. Sex, age, and marital status data were used to categorize each person censused. Those who did not belong to the four categories of interest (i.e., males, females under age 18) were not eligible. For each village, lists of eligible participants were created by category. Participants were then randomly selected from each list with the aim of including six participants per category per village.

#### 2.5.3. Phase 2, Stage 3: Survey Administration

The survey was administered by the team of enumerators that piloted the items (December 2014–February 2015). Data collected included participant demographics, water and sanitation access, Sanitation and Menstrual Insecurity items, water and sanitation behaviors, menstruation behaviors, social support, and mental health outcomes. Women who self-classified themselves as currently experiencing menstruation were eligible to answer menstruation-related questions and have responses included in the current analysis. Enumerators were advised to engage only one woman per household. Enumerators read questions to the participants in Oriya and marked the responses provided on a paper survey, which took approximately one hour.

### 2.6. Phase 3: Measurement Finalization

Measurement finalization followed steps outlined in previous work [35]. We assessed the overall sample then carried out Exploratory (EFA) and Confirmatory Factor Analysis (CFA). We then assessed measure reliability and validity.

#### 2.6.1. Phase 3, Stage 1: Final Sample Assessment and Creation of Random Sub-Samples

Participants were excluded from analysis if under age 18 and mistakenly sampled or had a household member previously surveyed (See Appendix A for Flow Diagram). Participant demographic information and relevant menstrual hygiene management practices were tallied by life stage to assess differences. We assessed the frequency, skewness, and kurtosis for each Menstrual Insecurity item (See Appendix A for item response frequencies, skewness, and kurtosis) [50].

The overall sample was randomly split into two sub-samples to carry out EFA (n_1_ = 426) and CFA (n_2_ = 452). We checked distributions, ran t-tests, and generated chi-square statistics to verify there were no significant differences in demographic and household information between the sub-samples (See Appendix A).

#### 2.6.2. Phase 3, Stage 2: Exploratory Factor Analysis

EFA, recommended as a first step in measurement development [50], was carried out to determine measure factor structure. Using Cattell’s scree test, a range for the number of optimal factors for the scale was determined. We assessed factor structures for models within this range to determine optimal fit [51].

Using the first sub-sample (n_1_), we carried out EFA on all 40 menstruation items using Mplus 8.3 software (Muthén and Muthén, Los Angeles, CA, USA). Given the hypothesized intercorrelated nature of the factors, an oblique rotation (QUARTIMIN) was used [50]. As survey responses were categorical, we used a mean and variance adjusted weighted least square (WLSMV) parameter estimator that uses chi-square test statistics to determine the final number of factors for the measure and factor loadings of each item. We ran models with the number of factors within the range identified by Cattell’s scree test (4–10 potential factors) and evaluated model fit statistics (root mean square error of approximation (RMSEA), comparative fit index (CFI), and Tucker-Lewis index (TLI), and theoretical fit to determine the final number of factors.

We assessed factor loadings and theoretical fit of each item within the factors to determine if an item should be dropped. We decided *a priori* to assess factor loadings based on cut-off points; a factor loading less than 0.3 indicated poor loading. Items were flagged as cross-loaded if the item loaded on two or more factors with a loading greater than 0.4 and the difference between the factor loadings was less than 0.3. First, we dropped poorly loaded items iteratively, dropping items with the lowest loadings first. Items that cross-loaded on multiple factors, or had the lowest loading below 0.6, were dropped iteratively [52]. The final factor structure was assessed using knowledge of women’s experiences with menstruation to ensure that items within factors and factors produced were appropriate and relevant. We used model fit statistics (noted above) to examine the final structure.

#### 2.6.3. Phase 3, Stage 3: Confirmatory Factor Analysis

We used the second sub-sample (n_2_) to test the factor structure identified through EFA. We used the Mplus 8.3 WLSMV estimator for CFA to determine the final model structure. The RMSEA, CFI, and TLI were used to examine the model fit. The same criteria as in the EFA was used to assess final factor loadings; any items below 0.3 were to be dropped from the model. Hierarchical confirmatory factor analysis was used to confirm that sub-scales form a unidimensional scale.

#### 2.6.4. Phase 3, Stage 4: Reliability and Validity Analysis

We analyzed the reliability (internal consistency) of the full scale and each individual factor by calculating Cronbach’s alpha using SAS 9.4 (SAS Institute, Cary, NC, USA). Pearson’s correlations between the full scored measure and each individual factor, as well as between each factor were used to assess potential redundancy of factors.

As previously noted, content validity was assessed by two external experts and the two Research Assistants (RAs) who led FLIs and FGDs, and face validity was assessed by both the RAs and the enumerator team. Convergent validity was determined by assessing correlations between each subscale score and items from the survey that asked about thematically similar concepts. Specifically, for Subscale 1 (Restrictions), we assessed correlation of scores with traveling alone; Subscale 2 (Management) scores with having resources to change menstruation material; Subscale 3 (Symptoms) scores with infection symptoms, and Subscale 4 (Menstrual Cycle Concerns) scores with experiencing regular menstruation. Additionally, known-groups validation was used to assess discriminant validity, with two sample t-tests being used to verify whether Subscale 5 (Menstruation-Related Bodily Concerns) scores differed by those reporting burning or itching in the vaginal area.

### 2.7. Menstrual Insecurity Scores

Sub-scale and overall scores were calculated as means (potential score ranges: 1.0–4.0). These numbers corresponded to the original response scale (1 = never, 2 = sometimes, 3 = often, 4 = always), with higher numbers indicating a greater mean frequency of occurrence. In addition, weighted scores were calculated, where standardization estimates from the variances of the sub-scales were used to weight each item before means were calculated.

Using SAS 9.4, we performed t-tests to determine if the mean scores were significantly different by life stage, ownership of a functional latrine, and type of menstruation material used. We also assessed differences in mean scores by responses to two items omitted from the scale. One assesses tension at menstrual onset, meaning a general feeling of anxiety about menstruation (not pre-menstrual stress or PMS, which can occur one to two weeks prior to menstruation), and the other assesses perceived difficulty doing regular work during menstruation. Analyses with these two items were not conceptualized *a priori* but were added after these items were omitted as a preliminary investigation of how these scores may vary by reported experiences of tension and work.

### 2.8. Ethics

Emory University’s Institutional Review Board (Atlanta, GA, USA; IRB00072840) and KIIT University’s Institutional Ethics Committee (Bhubaneswar, India; KIMS/KIIT/IEC/795/2014) granted ethical approval for this research. Participants provided informed oral consent prior to engagement.

## 3. Results

### 3.1. Final Sample Assessment

#### 3.1.1. Overall Sample

We administered 1437 surveys; 29 were excluded because participants were under age 18 or another family member participated. Of the remaining 1408, the mean age of menarche for the total sample was 14.71 years-old; 17 (1.2%) did not know and 3 (0.2%) reported never having experienced menstruation. A total of 879 (62%) reported they were currently experiencing menstruation and were eligible to answer questions in the menstruation module; 750 (85%) self-reported regular menstruation and 129 (15%) self-reported irregular menstruation. Of the 524 (37%) who were not experiencing menstruation and did not complete the module, 76 (14%) were pregnant, 45 (9%) recently gave birth, and 403 (77%) were menopausal. One respondent had missing data, so final measurement sample size was 878, including 338 (39%) unmarried women (stage 1), 222 (25%) women married for three years or less (stage 2), 308 (35%) women married over three years, and 10 (<1%) women over age 49 of any marital status (Table 1). (See Appendix A for Flow Diagram).

In the analytic sample, the average age of participants was 27, and the majority were Hindu (866; 99%), had at least some schooling (809; 92%), and had a “below the poverty line” (BPL) card providing them access to government support (549; 63%). Only 33% (288) had access to a functional household latrine, and fewer had a primary drinking source (242; 29%) or an enclosed bathing space within their household compound (131; 15%). The majority were General Caste (392; 45%) and relied exclusively on reusable cloth during menstruation (611; 70%) (Table 1).

Most women changed their materials in a private room in their home (782; 90%). Among women who used disposable materials, almost all disposed of pads in a pond/river/stream (172; 66%) or an outside trash pile (70; 27%). Among women that reused materials, the majority washed cloths in a pond/river (434; 71%) and dried them outside in the sun (487; 80%). Women stored reusable materials hidden in their home (303; 50%) or in the eaves of their house roof (247; 41%) (See Appendix A for menstruation-related behaviors).

#### 3.1.2. Menstruation Module Items

The most commonly reported concerns were not related to management. Specifically, women reported at least sometimes worrying about not being able to participate in religious activities during menstruation (84%); not being able to touch things during menstruation, which caused difficulties (66%); experiencing a general feeling of tension at menstrual onset (66%); experiencing stomach pains during menstruation (61%); avoiding leaving the house during menstruation (46%); and having difficulty doing their regular work during menstruation (43%) (Table 2).

Participants most frequently responded ‘always’ to questions regarding physical symptoms during menstruation or limitations to their activities during menstruation, such as: “Experienced stomach pains during menstruation” (29%) and “Worry about not being able to participate in religious activities” (29%). Women most frequently responded ‘never’ to questions related to social concerns: “Worried that others knew I was menstruating” (98%) and “Had people tease me because they knew I was menstruating” (99%).

#### 3.1.3. Split Samples Created for Analysis

Random splits resulted in 426 observations for EFA and 452 observations for CFA. T-tests to compare the groups revealed an even split, with comparable distributions in each group (See Appendix A for split group demographics) and were deemed similar to obtain valid results for EFA and CFA. All but two menstruation module items had similar response distributions in the two groups; ‘Had to stay separated at night from my normal bed’ and ‘Had to wear a cloth that was not fully dry after washing’ had different response proportions between the groups (See Appendix A for split group item response frequencies).

### 3.2. Exploratory Factor Analysis

The initial scree plot of the data indicated that a 7-factor structure may provide the best fit. Based on this information and our original hypothesis that there would be four factors linked to the identified themes, we assessed models from 4 to 10 factors to determine which was best-suited. We considered both model fit statistics and overall theoretical fit of the items to the factors. Using the Mplus 8.3 QUARTMIN rotation, we determined the 5-factor solution provided the best balance between theoretical fit and model statistics. After iteratively dropping items based on factor loading values and cross-loadings, 19 of the 40 items were included in the final 5-factor scale, which explained 69% of the variance among those 19 items (above the 60% threshold [53]) (See Appendix A for reasons items were dropped). The model produced strong, positive loadings and had good model fit statistics (RMSEA = 0.027; CFI = 0.994; TLI = 0.989; Table 3).

Based on the items within, we labeled the five factors: “Restrictions,” “Management”, “Symptoms,” “Menstrual Cycle Concerns,” and “Menstruation-Related Bodily Concerns” (referred to moving forward as ‘Bodily Concerns’ for ease). Factor 1 (Restrictions) includes two items about concerns continuing regular activities during menstruation (factor loadings: 0.866–0.998; 9.4% variance explained). Factor 2 (Management) contains nine items about women’s concerns about their ability to access, wash, and store their chosen materials and care for themselves and their menstrual needs during menstruation (factor loadings: 0.482–0.932; 27.1% variance explained). Factor 3 (Symptoms) contains two items about physical symptoms experienced during menstruation (factor loadings: 0.847–0.883; 8.3% variance explained). Factor 4 (Menstrual Cycle Concerns) contains three items about women’s concerns about their menstrual cycle or overall health related to their menstruation (factor loadings: 0.800–1.035; 12.7% variance explained). Finally, Factor 5 (Bodily Concerns) includes three items that address the concerns or experiences related to the body when using menstruation materials (factor loadings: 0.580–1.058; 11.6% variance explained). As hypothesized, the ‘Management’ and ‘Restrictions’ factors were retained. Our hypothesized ‘Well-Being’ factor essentially became two factors, Symptoms and Menstrual Cycle Concerns. The Bodily Concerns factor emerged to include items from three of the hypothesized factors.

### 3.3. Confirmatory Factor Analysis

All items loaded in the CFA in similar ranges as for the EFA and were significant (Table 3). No additional items were dropped. The model fit statistics provided satisfactory evidence that the factor structure was appropriate for the data (RMSEA = 0.058; CFI = 0.937; TLI = 0.925). Hierarchical CFA confirmed that all sub-scales form a unidimensional scale, and that a total score can be used to assess menstrual insecurity (RMSEA = 0.057; CFI = 0.936; TLI = 0.926; Appendix A for Hierarchical CFA Model Diagram). Additionally, bifactor CFA models with and without factor 3 were used to confirm its psychometric relevance to the overall factor structure, despite having the lowest factor variance in EFA.

### 3.4. Reliability and Validity Analyses

#### 3.4.1. Reliability Analysis

Cronbach’s alpha for the full 19-item scale was 0.75, which exceeds the threshold for acceptable internal consistency-reliability (Table 4). The items within each factor indicated moderate to high internal consistency based on the alpha values of each individual factor (0.63–0.86). All factors had low to moderate inter-correlations, reiterating that each factor is representing a related, but distinct component of Menstrual Insecurity.

#### 3.4.2. Validity

All factors were significantly associated with many thematically similar survey items (See Appendix A for items used for validation). The Restrictions factor was correlated with women’s ability to travel alone to locations outside of the village, the place where they defecate, and the place where they urinate (*p*s < 0.005). The Management factor was correlated with difficulty finding a private place to urinate and having an enclosed household bathing area (*p*s < 0.0001). The Symptoms factor was associated with having vaginal discharge, burning or itching while urinating, and burning or itching in the vaginal area in the previous two weeks (*p*s < 0.0001). The Menstrual Cycle Concerns factor was associated with self-reports of currently experiencing regular versus irregular menstruation (*p* < 0.0001) and the Bodily Concerns factor was associated with reusing absorbent material (vs. throwing away) (*p* < 0.005). Additionally, discriminant validity of the Bodily Concerns factor is supported by known-group validity. The score was significantly higher (*t* = 2.19, *p* < 0.05) among those who reported burning or itching in the vaginal area (*n* = 107) than among those who did not (*n* = 769).

#### 3.4.3. Menstrual Insecurity Scores

The mean total Menstrual Insecurity score was 1.46; scores were significantly higher (indicating greater insecurity) among those with no functional household latrine, no enclosed household bathing area, no within compound water source, and among those using reusable cloth materials.

Women who reported at least sometimes experiencing tension at menstrual onset had significantly higher Menstrual Insecurity scores than women who reported never experiencing tension (1.30). Those indicating they always experienced tension at menstrual onset reported the highest scores for Menstrual Insecurity overall (1.63), and for four of the five factors. Women who reported at least sometimes experiencing difficulty doing their regular work during menstruation had significantly higher Menstrual Insecurity scores than women who reported never having difficulty. Those indicating they always experienced difficulty reported the highest scores for Menstrual Insecurity overall (1.93), and for each of the five factors.

Mean scores for the full sample ranged from 1.15 (Factor 4: Menstrual Cycle Concerns) to 2.33 (Factor 1: Restrictions and Factor 3: Symptoms) (See Table 5 for Scores and Appendix A for Weighted Scores). While there was no difference in overall scores by life stage, differences did exist for select factors. Specifically, recently married women and married women had significantly higher scores for restrictions (RM = 2.38; M = 2.55) compared to unmarried women (UM = 2.07), and married women had a significantly lower score for symptoms (M = 2.11) compared to unmarried women (UM = 2.46).

Women who did not own a functional household latrine had higher scores across all five factors compared to women who did; scores were significantly higher for the restrictions (2.40) and management factors (1.29). Women who did not own an enclosed bathing area in their household had higher scores for four factors compared to women who did; scores were significantly higher for the restrictions (2.36) and management factors (1.27). Women who did not have a water source within their household compound had higher scores across four factors compared to women who had their water source outside the compound; scores were significantly higher for the management factor (1.29). Women who used a disposable pad during menstruation had significantly lower scores across four of the five factors compared to women who only used a reusable cloth; scores were significantly lower for the Management (1.17), Menstrual Cycle Concerns (1.09), and Bodily Concerns (1.17) factors.

## 4. Discussion

We used a rigorous, mixed methods approach to create and validate a culturally-grounded measure of Menstrual Insecurity, which we defined preliminarily as ‘the suite of social, environmental, and biological concerns and negative experiences resulting from menstruation’. The measure was designed to reflect the menstruation-related concerns and negative experiences of women from rural Odisha, India, and to assess the frequency of those concerns and negative experiences among a representative sample. Through our qualitative phase, we identified four themes that we hypothesized would emerge as factors that reflect conditions of women’s social and physical environments, and their individual experiences of menstruation, extending beyond but including management. Five factors emerged that deviate slightly from the four hypothesized, but are theoretically plausible and do reflect the social environment (Restrictions), the social and physical environment collectively (Management), and individual experience (Symptoms; Menstrual Cycle Concerns; Bodily Concerns) as noted in the definition. The measure is internally consistent, and sub-scale scores were significantly associated with thematically similar survey items, demonstrating validity. Post-hoc exploration of Menstrual Insecurity scores were highest among those reporting tension at menstrual onset or difficulty doing regular work during menstruation, and suggest the need for further research.

Our post-hoc finding that those experiencing tension at menstrual onset had significantly higher Menstrual Insecurity scores is consistent with qualitative studies [8,54,55,56]. For clarity, in this context tension is understood to be distress about menstruation, not emotional variations due to hormones, similar to what has been described by Weaver (2017) [57]. A pair of research studies with women in rural, urban, and tribal areas of Odisha identified menstruation to be a stressful sanitation-related activity [10] and the activity most likely to be ranked ‘most stressful’ [11]. The present research moves beyond the sanitation-specific lens to include aspects of menstruation beyond management. Items included in the final measure ask about restrictions, pain, and overall concerns about the menstrual cycle, among others, which have been noted to be under documented in current research, including in the foundational definition of menstrual hygiene management [23,58]. In our study, women reporting tension at menstrual onset had significantly higher Menstrual Insecurity scores overall, and for select sub-scales, suggesting the importance of these menstruation experiences and the need for further research exploring the relationship between the varied menstruation experiences and validated mental health outcomes.

We found that those who reported difficulty doing regular work during menstruation had significantly higher overall Menstrual Insecurity and individual sub-scale scores than those reporting they never faced difficulties. Research on menstruation and work is limited [59]. The highest scores reported were for the Symptoms and Restrictions factors, potential priority areas for future research and programming. We intentionally left our question open, allowing women to consider ‘regular work’ in their context, whether unpaid in the home or for income generation. Still, our analysis is limited to known groups and further research is needed to understand how menstruation impacts work, what types of work are impacted, the consequences of having difficulty engaging work, and what can be done to ameliorate menstruation-related work challenges.

There has been limited research focus on menstrual discomfort and pain, including impacts on menstruator’s lives, and amelioration strategies [23,58]. Our Symptoms and Bodily Concerns factors both address the bodily experience of menstruation; the Symptoms factor focuses on pain attributable to the menstrual cycle and the Bodily Concerns factor reflects women’s concerns about the body related to management, including wounds and smell. Ignoring pain is not new; research has documented how women’s reports of pain and discomfort for other health ailments have been disregarded and untreated [60]. Our research shows that menstrual pain and discomfort are a concern for women, associated with feelings of tension and difficulty working, can be from different causes (cycle or management), and need attention.

While the Symptoms and Bodily Concerns sub-scales do provide some insight, they also have limitations. The symptoms factor only has two items, limiting its comprehensiveness, and does not enable assessment of whether or not these symptoms are concerning for women, but just the frequency of occurrence. In future work, we would aim to revise this sub-scale by specifically asking women if they were concerned about a more expansive list of symptoms. Further, we think it also would be useful to test a culturally appropriate ‘Menstruation Symptoms Index’ to accompany this sub-scale. The index could be used to determine which, of a range of symptoms, women experience, and how intense the symptoms are. Some older measures do exist, and these would need testing to assure contextual appropriateness [61,62]. Similarly, we would revise items in the Bodily Concerns sub-scale to more explicitly assess worry or challenges related to the body as a result of management, rather than simply asking about an experience (e.g., concern about wounds from cloth).

Women who reported always having difficulty working during menstruation or feeling tension at menstrual onset had high restrictions scores, highlighting the need for further research to understand how menstruation-related restrictions impact menstruator’s lives. Van Eijk et al.’s systematic review of menstruation among girls in India noted myriad restrictions [63], and in their research on girls’ menstruation and schooling in Bangladesh, Alam et al. found school absence to be more common among girls who faced restrictions during menstruation. This factor needs further development and is likely to be quite varied contextually.

Women had significantly higher scores for Management if they lacked access to enabling WASH environments or used reusable cloth. Our quantitative findings that women have greater difficulty managing menstruation when they have limited WASH and absorbent material access are consistent with numerous qualitative studies that have amplified women’s and girls’ voiced challenges managing menstruation in schools and at home [10,12,22,64,65,66]. In this factor, one item may seem like an outlier (Had to stay separated at night from my normal bed). However, in the qualitative work women described sleeping out of their beds before and during menstruation not because of any restriction, but to avoid having to wash bed linens, and is therefore management-related. In our study, women had limited access to functional latrine facilities (33%), enclosed bathing spaces (15%), water sources within their compounds (29%), and disposable pads (23%). Despite better scores among those with WASH access, only a limited number of women used WASH facilities for changing (3% in latrine; 6% in bathing space) or washing reusable materials (4% in latrine; 4% in bathing space). Thus, access to functional WASH facilities are clearly of use to some women, but only among a minority. The majority of women changed in the house (89%) and used a pond or river for washing (71%) and disposing (66%) of materials. Other studies with women in India and Nigeria found access to and use of sanitation facilities for changing to be higher [24,40]. Low latrine use for menstrual needs among this rural Indian population is likely related to the well-documented aversion to using latrines even for urination and defecation [10,22,67,68] and because facilities are not designed to be ‘female friendly’ [69]. While women using pads do not have to find places to wash and dry materials, they struggle with how and where to privately dispose [12]. Our research found that almost all women in our sample were disposing materials directly into the environment. Urgent attention to environmentally-forward material and disposal solutions are needed, particularly as global markets and subsidies for disposables continue to grow [70,71].

The Menstrual Cycle Concerns sub-scale highlights the need women have for information about the menstrual cycle and their own health. In the preceding qualitative research, participants asked research assistants if something was wrong with their health if their menstruation came on different dates each month [12]. That women were asking RAs about their cycles demonstrates that women have few sources of information about menstruation or health. Many studies have concluded that girls need to understand the basic biology of menstruation and links to reproduction [8,72], but it is clear that women also need information. Women’s access to accurate information can benefit their own lives and also girls. Women are the mothers, sisters, grandmothers, and aunts who provide menstruation information to adolescents [72]. Women who have access to disposable pads had significantly lower scores for the Menstrual Cycle Concerns sub-scale. It may be that those with access to disposable products may also have better access to other resources, like health care or information (booklets with menstruation information are often provided in packages of commercial menstrual products) and thus do not have as many unanswered questions or concerns as those not using pads regularly.

All the retained items in the Menstrual Cycle Concerns sub-scale had over 90% of women respond that they ‘never’ had the concern or experience. Comparatively, 27% of qualitative participants perceived irregular menstruation to be a concern for women in general [22]. While a high proportion of ‘never’ responses for these and other items may seem at odds with our qualitative work, which showed higher endorsement, differences were expected. The current measure assesses concerns at the individual level, whereas the qualitative work asked about perceived concerns among women in the community in general, a deliberate approach used to capture a range of perceived concerns and to enable participants to feel more comfortable talking about challenging topics. Our quantitative assessment demonstrates that, though concerns or experiences may be widely recognized qualitatively in a community, many are not or are infrequently experienced at an individual level, further highlighting the value of mixed methods assessments.

Similarly, for four items in the management factor, 90% or more of respondents indicated that they ‘never’ had the concern. Each of the four items queried about cloth or pads, specifically difficulty in accessing materials needed, or difficulty finding a place to dispose, change, or store materials. Given that a high proportion (70%) of the sample reported using reusable materials, it is not surprising that disposal was never an issue for 90% of respondents. For the other items, we simply asked about difficulty in accessing materials needed or in finding places for changing and storing in general. These items may have been too broad, too general. In the qualitative work, women discussed not being able to get materials they preferred or changing or storing materials in places they felt were not ideal. In future iterations of the scale, we would recommend refining these items to ask about perceived suitability of the materials accessed or places used for management. Women are resilient and will find a means to address their needs, but they may not do so in a manner that aligns with their preferences.

Finally, a number of items were not retained in our measure (See Appendix A), however many provide insightful information and can be used in future research even if not in the scale. Notably, during menstruation, many women experienced constrained mobility (24% were prevented from going to places they wanted to go to and 46% avoided leaving the house during menstruation), had difficulty accessing resources (37% had trouble fetching water), and had challenges meeting basic needs (29% had difficulty urinating).

### Strengths and Limitations

A strength of this research is the ground-up approach. Items emerged from qualitative work with rural Indian women of varied life stages, allowing their voices to shape the measure, and leading to a scale with factors that represent their concerns. We would have liked to conduct cognitive interviews with women to enable additional feedback, however resources were constrained and we were unable to do so. However, we did carry out a careful review process with the enumerator team members, who are from similar villages as the participants, which provided some additional insights regarding translation. The ground-up approach is also a limitation. Some items may not be relevant in other contexts and the concerns of women in other settings may not be represented. However, the factors, and many items, align with menstruators’ experiences across settings. Testing the scale, or even specific factors, in other settings and with other populations is needed. Additionally, our scale identified factors important to the menstruation experience beyond management. Two factors, Restrictions and Symptoms, have only two items and future work should develop these factors. It may be useful for each of these factors to be developed further as independent scales, depending on context, what may be of greatest utility to researchers and practitioners, and future learning in the field.

Due to constrained time and resources, we were not able to return to a subsample of participants to re-administer the survey and assess test-re-test reliability, nor were we able to carry out data collection at varied time points to assess predictive validity. Given that there are no comparable measures, we also were not able to assess criterion validity.

Finally, as research on menstruation evolves, the definition of Menstrual Insecurity, which is currently broad, may also evolve. Any future development of Menstrual Insecurity as a concept will have implications for the content validity of the current measure. In other words, the current items and sub-scales may not fully represent a future iteration of Menstrual Insecurity. Still, we consider this work and this measure to be a starting point and accept that the concept may evolve and that the measure may need to evolve as well. Indeed, concepts of food and water insecurity have changed over time, and so too have their measures, and those fields have benefited from those evolutions.

## 5. Conclusions

We created and validated a scale to measure Menstrual Insecurity, employing a rigorous, mixed-methods approach to reflect the concerns of women in rural Odisha India. The scale includes Management, Restrictions, Symptoms, Bodily Concerns, and Menstrual Cycle Concerns sub-scales, broadening what has to date been measured in menstruation-related research. We found that women who reported experiencing tension at menstrual onset and difficulty doing regular work to have higher Menstrual Insecurity scores. This measure may be used to identify needs for program planning. Future work can explore if there is an association between Menstrual Insecurity and health outcomes.

## Figures and Tables

**Figure 1 ijerph-17-03468-f001:**
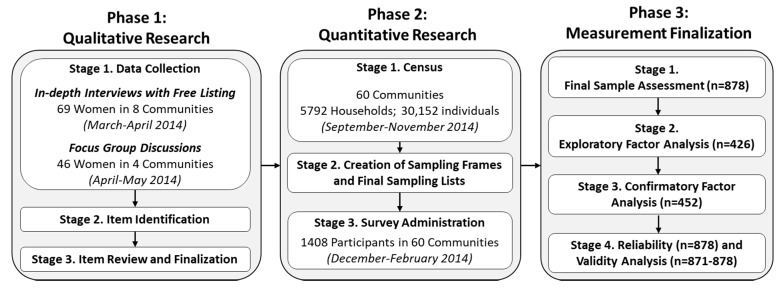
Schematic of Exploratory Sequential Mixed Methods Design Used to Create *Menstrual Insecurity* Measure.

**Table 1 ijerph-17-03468-t001:** Demographic characteristics of participants, overall and by life stage, in rural Orissa, India.

Characteristics	All		1. Unmarried (UM)		2. Recently Married (RM)		3. Married (M)		4. Over 49 (OW)	
**Total Surveyed ^1^**	1437		341	24.2%	320	22.7%	395	28.1%	352	25.0%
**Mean Age at First Menstruation**	14.71	10.22	13.66	4.79	13.65	5.02	14.87	10.58	16.49	15.57
Don’t Know	17	1.2%	0	0.0%	1	0.3%	6	1.5%	10	2.8%
Never Experienced Menstruation	3	0.2%	1	0.3%	0	0.0%	0	0.0%	2	0.6%
**Currently Experiencing Menstruation ^1^**										
Yes, self-reported regular menstruation	750	53.3%	302	88.8%	189	59.1%	254	64.3%	5	1.4%
Yes, self-reported irregular menstruation	129	9.2%	37	10.9%	33	10.3%	54	13.7%	5	1.4%
No, Pregnant	76	5.4%	1	0.3%	65	20.3%	10	2.5%	0	0.0%
No, Recently Gave Birth	45	3.2%	0	0.0%	32	10.0%	13	3.3%	0	0.0%
No, Menopause	403	28.6%	0	0.0%	1	0.3%	64	16.2%	338	97.1%
**Total Participants in** **Menstruation Sub-Study**	878		338	38.5%	222	25.3%	308	35.1%	10	1.1%
Former Intervention Community	436	49.7%	165	48.8%	104	46.8%	159	51.6%	8	80.0%
Age	26.80	7.68	21.22	2.95	24.01	3.06	34.09	6.05	52.70	4.00
Hindu	866	98.6%	336	99.4%	219	98.6%	301	97.7%	10	100.0%
Possession of Below Poverty Line (BPL) Card ^1^	549	62.6%	226	66.9%	137	61.7%	179	58.3%	7	70.0%
Have Children	447	50.9%	0	0.0%	140	63.1%	297	96.4%	10	100.0%
Mean Number of Children	1.01	1.26	0	0	0.68	0.56	2.30	1.13	3.40	0.70
No Current Illness ^1^	776	88.6%	306	90.5%	209	94.1%	258	84.3%	2	20.0%
**Caste**										
Brahmin	26	3.0%	10	3.0%	7	3.2%	9	2.9%	0	0.0%
General Caste	392	44.7%	149	44.1%	104	47.1%	135	43.8%	4	40.0%
Scheduled Caste (SC)	158	18.0%	51	15.1%	43	19.5%	62	20.1%	2	20.0%
Other Backward Caste (OBC)	283	32.3%	122	36.1%	60	27.1%	97	31.5%	4	40.0%
Scheduled Tribe (ST)	5	0.6%	2	0.6%	2	0.9%	1	0.3%	0	0.0%
Don’t Know	12	1.4%	3	0.9%	5	2.3%	4	1.3%	0	0.0%
**Education**										
None	69	7.9%	3	0.9%	6	2.7%	55	17.9%	5	50.0%
Some Primary	245	27.9%	53	15.7%	44	19.8%	143	46.4%	5	50.0%
Some Secondary	492	56.0%	232	68.6%	161	72.5%	99	32.1%	0	0.0%
Higher than Secondary	72	8.2%	50	14.8%	11	5.0%	11	3.6%	0	0.0%
**Absorbent Material** **Used for Menstruation ^1^**										
Reusable Cloth	611	69.6%	224	66.3%	112	50.5%	265	86.9%	10	100.0%
Disposable Pad	204	23.2%	83	24.6%	89	40.1%	31	10.2%	0	0.0%
Both Cloth and Pad	61	6.9%	31	9.2%	21	9.5%	9	3.0%	0	0.0%
**Household Water and Sanitation Access**										
Functional Latrine in Household ^1^	288	32.9%	94	27.8%	101	45.5%	89	29.1%	4	40.0%
Primary Drinking Water Source within Dwelling/Compound ^2^	242	29.3%	83	25.9%	74	36.3%	81	27.8%	4	40.0%
Bathing Room in Household ^1^	131	15.1%	26	7.7%	66	29.9%	38	12.5%	1	10.0%

Data are number and percent or mean (and standard deviation). ^1^ For *Total Surveyed:* 29 excluded because family member surveyed or under age 18; For *Currently Experiencing Menstruation: 5* missing (stage 1 & stage 4); For *Caste*: 2 missing (stage 1 & stage 2) and 12 indicated don’t know; For *BPL Card*: 1 missing (stage 3); For *No Current Illness*: 2 missing (stage 3); For *Absorbent Material Used:* 1 missing (stage 3); For *Bathing Room in Household*: 8 missing (stage 1, stage2, & stage 3). ^2^ For *Primary Drinking Source*: data taken from census, 53 missing (stage 1, stage 2, & stage 3); For *Functional Latrine in Household*: data taken from census, 2 missing (stage 3).

**Table 2 ijerph-17-03468-t002:** Responses frequencies for all *Menstrual Insecurity* items among women in rural Orissa, India, organized by factor analysis results and including deleted items (n = 878).

	Full Sample (%)			
	Never	Some-times	Often	Always
**Factor 1: Restrictions**				
Could not touch certain things, which created difficulties for me	297 (34)	276 (31)	129 (15)	176 (20)
Worry about not being able to participate in religious activities	241 (27)	261 (30)	121 (25)	255 (29)
**Factor 2: Management**				
Had difficulty finding a place to change menstrual materials (cloth or pad)	804 (92)	35 (4)	15 (2)	24 (3)
Had difficulty finding a place to wash cloth	717 (82)	51 (6)	40 (5)	70 (8)
Had difficulty finding a place to store menstrual cloth or pads	819 (93)	30 (3)	9 (1)	20 (2)
Experienced difficulty bathing during menstruation	676 (77)	103 (12)	43 (5)	56 (6)
Had difficulty finding a place to dispose of cloth or pad	791 (90)	28 (3)	22 (3)	37 (4)
Could not get the material I needed, like cotton cloth or sanitary pad	832 (95)	33 (4)	8 (1)	5 (1)
Had difficulty finding a suitable place to dry my menstrual cloth	749 (85)	49 (6)	27 (3)	53 (6)
Had difficulty finding someone to help me with bathing and other menstruation-related needs at onset	683 (78)	152 (17)	17 (2)	26 (3)
Had to stay separated at night from my normal bed	762 (87)	69 (8)	13 (1)	34 (4)
**Factor 3: Symptoms**				
Experienced pain in the hands or legs during menstruation	329 (37)	174 (20)	134 (15)	241 (27)
Experienced stomach pains during menstruation	340 (39)	179 (20)	105 (12)	254 (29)
**Factor 4: Menstrual Cycle Concerns**				
Worried that my cycle was irregular	798 (91)	26 (3)	10 (1)	44 (5)
Worried about ability to become pregnant because of problems with my menstrual cycle	849 (97)	12 (1)	7 (1)	10 (1)
Worried about my health because of problems with my menstrual cycle	790 (90)	44 (5)	12 (1)	32 (4)
**Factor 5: Bodily Concerns**				
Got wounds on inner thighs from cloth or belt	655 (75)	182 (21)	20 (2)	21 (2)
Worried that my cloth, napkin or my body smelled	756 (86)	42 (5)	42 (5)	38 (4)
Had trouble doing my work because of wound from cloth/pad	774 (88)	74 (8)	11 (1)	19 (2)
**Items Deleted Based On Factor Analysis**				
Avoided leaving the house during menstruation	470 (54)	53 (6)	162 (18)	193 (22)
Did not feel like eating during menstruation	471 (54)	227 (26)	77 (9)	103 (12)
Worried about needing other’s support to get the cloth or sanitary pads I needed	688 (78)	140 (16)	25 (3)	25 (3)
Experienced headache during menstruation	693 (79)	114 (13)	33 (4)	38 (4)
Did not feel like interacting with others during menstruation	668 (76)	112 (13)	33 (4)	65 (7)
Experienced burning and irritation in urinary tract during menstruation	683 (78)	131 (15)	36 (4)	28 (3)
Got blood stains or leaks on my clothes	755 (86)	106 (12)	11 (1)	6 (1)
Experienced heavy bleeding	712 (81)	98 (11)	34 (4)	34 (4)
Felt like vomiting during menstruation	779 (89)	50 (6)	19 (2)	30 (3)
Had a lot of cleaning work to do because of menstruation	627 (71)	87 (10)	113 (13)	51 (6)
Had difficulty doing my regular work during menstruation	506 (58)	190 (22)	105 (12)	77 (9)
Had difficulty fetching water for menstruation related needs	556 (63)	143 (16)	88 (10)	91 (10)
Had difficulty urinating while menstruating	627 (71)	147 (17)	50 (6)	54 (6)
Had difficulty walking during menstruation	508 (58)	230 (26)	53 (6)	87 (10)
Was forced to conform to a restriction that I do not believe in	729 (83)	117 (13)	9 (1)	23 (3)
Had people tease me because they knew I was menstruating	865 (99)	11 (1)	1 (0)	1 (0)
Worried about being treated as untouchable by others	832 (95)	26 (3)	6 (1)	14 (2)
Had to wear cloth that was not fully dry after washing *	847 (97)	26 (3)	4 (0)	0 (0)
Have a general feeling of tension at onset of menstruation	307 (35)	139 (16)	180 (21)	252 (29)
Was prevented from going to certain places that I wanted to go	666 (76)	110 (13)	11 (1)	91 (10)
Worried that others knew I was menstruating	862 (98)	9 (1)	3 (0)	4 (0)

* 1 participant missing (Sub-sample n_2_ = 451).

**Table 3 ijerph-17-03468-t003:** Final factor loadings and model fit statistics for random split-half sample EFA (N1 = 426) and CFA models (N2 = 452).

Model	Item Code	EFA (N_1_ = 426)	CFA ^1^ (N_2_ = 452)
**Factor 1: Restrictions**			
Could not touch certain things, which created difficulties for me	M14	0.866	0.847 *
Worry about not being able to participate in religious activities	M13	0.998	0.975 *
**Factor 2: Management**			
Had difficulty finding a place to change menstrual materials (cloth or pad)	M29	0.836	0.911 *
Had difficulty finding a place to wash cloth	M21	0.931	0.908 *
Had difficulty finding a place to store menstrual cloth or pads	M39	0.813	0.826 *
Experienced difficulty bathing during menstruation	M19	0.672	0.787 *
Had difficulty finding a place to dispose of cloth or pad	M33	0.820	0.736 *
Could not get the material I needed, like cotton cloth or sanitary pad	M11	0.606	0.485 *
Had difficulty finding a suitable place to dry my menstrual cloth	M25	0.932	0.763 *
Had difficulty finding someone to help me with bathing and other menstruation-related needs at onset	M38	0.645	0.629 *
Had to stay separated at night from my normal bed	M40	0.482	0.326 *
**Factor 3: Symptoms**			
Experienced pain in the hands or legs during menstruation	M23	0.883	1.074 *
Experienced stomach pains during menstruation	M24	0.847	0.689 *
**Factor 4: Menstrual Cycle Concerns**			
Worried that my cycle was irregular	M37	1.035	0.835 *
Worried about ability to become pregnant because of problems with my menstrual cycle	M47	0.815	0.852 *
Worried about my health because of problems with my menstrual cycle	M41	0.800	0.871 *
**Factor 5: Bodily Concerns**			
Got wounds on inner thighs from cloth or belt	M28	0.808	0.628 *
Worried that my cloth, napkin or my body smelled	M48	0.580	0.401 *
Had trouble doing my work because of wound from cloth/pad	M44	1.058	1.315 *
**Model Fit Statistics**			
RMSEA		0.027	0.058
CFI		0.994	0.937
TLI		0.989	0.925

^1^ CFA Factor Loadings standardized using STDYX Standardization in MPLUS; * *p* ≤ 0.01.

**Table 4 ijerph-17-03468-t004:** Scale and Factor descriptive characteristics, inter-correlations, and internal consistency reliability.

	No. of Items	Mean Score (SD)	Correlations between Scale Factors					Internal Consistency Reliability ^1^
			F1	F2	F3	F4	F5	
**Total Score**	19	1.46 (0.34)	0.57 *	0.79 *	0.55 *	0.36 *	0.38 *	0.75
F1: Restrictions	2	2.33 (1.07)		0.23 *	0.16 *	0.05	0.16 *	0.86
F2: Management	9	1.24 (0.42)			0.20 *	0.13 *	0.13 *	0.80
F3: Symptoms	2	2.32 (1.12)				0.13 *	0.03	0.77
F4: Menstrual Cycle Concerns	3	1.15 (0.46)					0.05	0.69
F5: Bodily Concerns	3	1.26 (0.49)						0.63

^1^ Cronbach’s alpha reported for internal consistency reliability. * *p* < 0.05.

**Table 5 ijerph-17-03468-t005:** Overall *Menstrual Insecurity* and individual sub-scale scores by life stage-, WASH-, and health-related items (N = 878).

Characteristics	Overall Menstrual Insecurity	Factor 1: Restrictions	Factor 2: Management	Factor 3: Symptoms	Factor 4: Menstrual Cycle Concerns	Factor 5: Bodily Concerns
**All**	1.46 (0.34)	2.32 (1.07)	1.24 (0.42)	2.32 (1.12)	1.15 (0.46)	1.26 (0.49)
**Life Stage**						
Unmarried Women (Ref.)	1.45 (0.34)	2.07 (0.99)	1.24 (0.41)	2.46 (1.12)	1.14 (0.43)	1.27 (0.51)
Recently Married Women	1.49 (0.38)	2.38 (1.06) *	1.27 (0.49)	2.42 (1.17)	1.18 (0.51)	1.21 (0.47)
Married Women	1.44 (0.32)	2.55 (1.11) *	1.21 (0.39)	2.11 (1.06) *	1.13 (0.45)	1.28 (0.49)
Older Women	1.44 (0.23)	2.70 (1.16)	1.17 (0.28)	1.95 (1.23)	1.27 (0.64)	1.27 (0.49)
**Own Functional Latrine**						
Owns (Ref.)	1.38 (0.28)	2.19 (1.05)	1.14 (0.30)	2.28 (1.09)	1.13 (0.43)	1.22 (0.46)
Does Not Own	1.49 (0.36) *	2.40 (1.08) *	1.29 (0.46) *	2.34 (1.14)	1.16 (0.47)	1.27 (0.51)
**Enclosed Bathing Area ^1^**						
Owns (Ref.)	1.34 (0.25)	2.14 (1.04)	1.08 (0.21)	2.32 (1.14)	1.09 (0.36)	1.18 (0.45)
Does Not Own	1.48 (0.35) *	2.36 (1.08) *	1.27 (0.45) *	2.32 (1.12)	1.16 (0.47)	1.27 (0.50)
**Water Source ^1^**						
Inside Compound (Ref.)	1.39 (0.27)	2.29 (1.06)	1.12 (0.26)	2.38 (1.12)	1.11 (0.40)	1.24 (0.49)
Outside Compound	1.48 (0.36) *	2.35 (1.10)	1.29 (0.46) *	2.30 (1.12)	1.16 (0.48)	1.26 (0.50)
**Menstruation Materials ^2^**						
Reusable Cloth (Ref.)	1.47 (0.34)	2.37 (1.08)	1.25 (0.42)	2.27 (1.11)	1.17 (0.49)	1.30 (0.53)
Disposable Pad	1.40 (0.29) *	2.22 (1.02)	1.17 (0.35) *	2.45 (1.17)	1.09 (0.31) *	1.17 (0.40) *
Both Pad and Cloth	1.54 (0.44)	2.34 (1.11)	1.40 (0.60)	2.43 (1.08)	1.21 (0.57)	1.16 (0.35) *
Nothing	1.05 (0.07)	1.00 (0.00) *	1.00 (0.00) *	1.25 (0.35)	1.17 (0.24)	1.00 (0.00) *
**Tension at Menstruation Onset**						
Never (Ref.)	1.30 (0.25)	2.00 (0.96)	1.12 (0.27)	2.01 (1.05)	1.13 (0.39)	1.10 (0.21)
Sometimes	1.47 (0.32) *	2.17 (0.88)	1.28 (0.43) *	2.35 (1.08) *	1.23 (0.55)	1.24 (0.37) *
Often	1.46 (0.26) *	2.36 (0.92) *	1.22 (0.32) *	2.25 (1.01) *	1.08 (0.31)	1.45 (0.58) *
Always	1.63 (0.41) *	2.78 (1.23) *	1.39 (0.57) *	2.73 (1.18) *	1.18 (0.56)	1.32 (0.65) *
**Had Difficulty Doing My Regular Work During Menstruation**						
Never (Ref.)	1.35 (0.27)	2.12 (1.03)	1.14 (0.29)	2.19 (1.12)	1.12 (0.39)	1.15 (0.35)
Sometimes	1.47 (0.29) *	2.31 (0.91) *	1.27 (0.40) *	2.33 (1.03)	1.15 (0.49)	1.29 (0.46) *
Often	1.58 (0.27) *	2.86 (1.00) *	1.28 (0.36) *	2.32 (1.11)	1.15 (0.45)	1.57 (0.71) *
Always	1.93 (0.49) *	3.04 (1.24) *	1.79 (0.75) *	3.10 (1.05) *	1.34 (0.70) *	1.44 (0.74) *

^1^ For *Enclosed Bathing Area:* 8 missing; For *Water Source*: 53 participants missing. ^2^ For *Menstruation Materials*: 1 participant missing, 611 Reusable Cloth, 203 Disposable Pad, 61 Both, 2 None; * *p*-value < 0.05.

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
