# Peer review of "Assessing Women’s Menstruation Concerns and Experiences in Rural India: Development and Validation of a Menstrual Insecurity Measure"

_ijerph, 2020, doi:10.3390/ijerph17103468_

Round 1

Reviewer 1 Report

This manuscript reports on the development of a ‘Menstruation Insecurity’ measure which collects data on an array of menstrual-related discomforts and difficulties within a single scale.

I do not believe this can be published in its current form, and significant revisions, including significant updates to the validation analyses would be required for this to be accepted.

I have three most significant concerns (the conceptual clarity of the work and framing of some of the sub-scales, the lack of endorsement of many scale items in the quantitative survey and thus disconnect with the qualitative development work, and the post-hoc selection of rejected scale items for validation), along with a collection of other questions/issues.

I have provided more information on my concerns related to these below.

  1. Conceptual clarity

Concept definition & fit with research and theory

As a reader, the concept of ‘menstruation insecurity’ was not clear. I could not understand the source of the definition, its relationship to other research in menstrual health and hygiene, theory, or the linked sanitation insecurity research.

Authors define menstrual insecurity as “when, as a result of menstruation, social, environmental and biological conditions constrain resources, capacity, confidence, agency, and dignity to care for the self and pursue the activities of daily life.” Notwithstanding that the sentence itself is hard to read, it isn’t clear if ‘menstrual insecurity’ is the social, environmental and biological constraints related to menstruation, or these constraints only when they impact capacity, confidence, agency and dignity (none of these concepts are represented in the scale items, certainly not confidence or agency). Or are these menstrual insecurity only if they also impact caring for the self and pursuing daily activities?

It reads as “any challenge impacting menstrual experience, any part of menstrual experience, and any outcomes of difficulty managing menstruation.” While this may fit with the authors approach of generating items from free listing interviews without imposing an external structure, it does not fit well with a theoretically cogent concept.

It is particularly confusing in the context of the Introduction. Authors do a fantastic job of introducing the need for quantitative measures to further the objectives of menstrual health research, they then go on to contextualize this need in light of measuring menstrual experiences using the framework presented in a recent systematic review of qualitative studies. On line 57 authors suggest that they will “develop a measure to assess menstruation experiences, which can be used independently and to determine how experiences influence health, employment or other impacts” – yet in their definition of menstrual insecurity authors clearly include impacts on everything from confidence and agency to employment and participation in daily life. The definition of menstrual insecurity covers from antecedents of menstrual experience (line 50), experience (line 51) and consequences on health and wellbeing (line 53).

How can the measure be used to assess the association between menstrual insecurity and impacts on health, employment or other impacts, when the concept of menstrual insecurity already includes those impacts? (items in the final scale include having to be separated from one’s usual bed – presumably engagement in normal daily life – an impact – and physical health harms – developing wounds due to cloth). Would it really be appropriate to test the association between menstrual insecurity and impacts if the menstrual insecurity measure already includes some of these impacts? Won’t we be generating false associations by conflating our predictor with our outcome? 

The definition of menstrual insecurity is also a curious fit with the authors definition of sanitation insecurity from 2017. In the same program of work, authors define sanitation insecurity as “insufficient and uncertain access to socio-cultural and social environments that respect and respond to the sanitation needs of individuals, and adequate physical spaces and resources for independently, comfortably, safely, hygienically, and privately urinating, defecating and managing menses with dignity at any time of day or year as needs arise in a manner that prevents fecal contamination of the environment and promotes health”  This definition seems much clearer.  Similar to menstrual insecurity this definition focuses on social and physical contributors, in this case to comfortable and safe sanitation behaviors. Unlike the definition of menstrual insecurity it doesn’t then extend to the consequences. It is more clearly focused on the range of antecedents and anxieties about lacking access to social and physical resources - whereas the definition of menstrual insecurity evokes a range of other concepts.

Researchers in menstrual health/hygiene can and should debate the merits of more comprehensive vs specific measures of concepts  - “lumping” vs “splitting” and the role of sub-scales vs alternative scales for measuring different concepts and it is likely different research and programmes would call for different approaches. I am somewhat uncomfortable with authors suggesting that this measure is the only thing expanding a focus away from menstrual management - this has been done across a range of comment and qualitative research - that these haven't been included in a single measure but across multiple measures isn't exactly a limitation. Toning this down would be appropriate.

Authors here argue their approach here is “more inclusive” but don’t clarify the boundaries of their measured concept and the relationship to other concepts in menstrual research (e.g., confidence/self-efficacy, social support). It is stated that menstrual insecurity includes social and physical environment antecedents, which it does include some, but it omits others (likely because these weren’t raised in the free listing interviews in this particular context) – these should be addressed at some point in the paper – there is limited info on social support or environmental infrastructure which we might think of immediately as social and physical antecedents of menstrual experience.

Authors should return to the concept definition in the discussion to further clarify the concept for measurement. It is not straightforward to understand so readers are going to need some help in how the factors and items identified accurately reflect menstrual insecurity. Especially when the definition includes concepts like confidence, agency and dignity, but the measure includes none of these concepts.

Sub-scale/factor labeling

Related to the concept, I am confused by some of the labels used for the factors/sub-scales. “Menstrual health” has been used much more comprehensively than only to refer to menstrual cycle irregularities (e.g., UNICEF 2019, FSG report). Many now refer to the study of menstrual health and include social and psychological health as part of health (as indicated by the WHO definition of health). Given the context of research in this field I would strongly advise against labeling this factor as ‘menstrual health’ – I think it would cause a lot of confusion. In the paper I wasn’t sure if authors were conceptualizing this factor as really capturing concerns about menstruation due to a lack of menstrual cycle knowledge, or if this factor actually captures women with disordered menstruation. Just as the overarching concept could do with more clarity, so too could the diverse factors included in the very broad scale.

I don’t think it is appropriate for factor 5 to be titled ‘consequences’. This again is likely to lead to misinterpretation that this captures consequences of menstrual experience for women’s lives. It captures one physical consequence of difficulties with materials – wounds, difficulties with work, only in relation to this wound, and worries about odour.

In methods authors state that a 5-factor structure was hypothesized. What were the a-priori hypotheses?

I think more clarity is needed on the 'symptoms' concept. Unlike the 'menstrual health' factor which is framed around concerns about irregularities - the 'symptoms' factor is just about if someone experiences menstrual pains. This seems like more of a biological characteristic. I'm not convinced of the value of including it in this single measure vs separately - but more than that - is this something that is going to respond to interventions? Absolutely we need to pay more attention to women's pain but even an intervention that provided better diagnosis and access to pain relief - would women answer this question differently? They would still 'experience' the onset of pain - it just might be less impacting or distressing because they have better management options. They might experience pain for less time (e.g., they take a pain-killer) but are the questions here going to be sensitive to that?

  1. The poor endorsement of many scale items, particularly the disconnect between the quantitative findings and what was identified through the qualitative process calls into question the validity of the scale and the authors strong claims about a ‘grounded’ and valid measure

The measure was developed through FGDs and free listing interviews. This grounded process is a strength of the study, and prioritizes women’s concerns in this context.

However, the lack of consistency between the concerns that seem to have been raised during the qualitative component, their weighting in exercises during development, and then the endorsement of items in the quantitative scale is highly concerning.

It would be helpful for readers to include in methods (ie. Lines 168-171) the thematic groupings of concerns as identified through the FGDs and FLIs – this could then be compared to the emergent factors. I don’t believe referral to supplementary materials in another paper is sufficient for the readership of this paper. How did the emergent themes from the qualitative study inform the concept of menstrual insecurity and the hypothesized factors for the scale, and the items for inclusion? Presumably the authors had some a-priori concepts they had developed and identified if these would be appropriate for measurement within a scale of menstrual insecurity (i.e., which concerns raised fit with the 'menstrual insecurity' concept). In results authors discuss balancing model fit with ‘theoretical fit’ of the data.

From the 2017 paper it looks like some of these groups were ‘bathing’ ‘washing cloth’ ‘drying cloth’ – yet none of these appear as factors (they do appear as items).

This information could be provided in preference to other chunks of text that aren’t that useful for readers of this study For example lines 113-121 could be significantly shortened – while it makes sense to contextualize this work in the context of others with the same program of work this is confusing for the reader and not essential information for this paper which already requires a lot of attention from the reader without further distractions. I’m not sure you need all the life-stage groupings in Table 1 – especially since there were few differences in the scale across – I think this would be more appropriate for supplementary materials and you could then have more scale relevant information in the main text.

From the 2017 paper, and the introduction of this paper it looks like women “reported 32 unique menstruation concerns". The most endorsed concerns were concerns about bathing (52%), washing (51%) and drying cloths (46%). And yet, in the quantitative survey and presentation of scale results, 82% of women reported never having trouble finding a place to wash their cloth, 77% reported no difficulties bathing, and 85% reported no issues finding a place to dry their cloth. This is never addressed in this paper or in the limitations.

I find it deeply concerning that the authors overlook this disparity between the qualitative findings and scale results. If this measure is truly grounded in the experiences of women in the study context, then how can the scale have such low endorsement of many items? In the introduction authors report the concerns raised in the qualitative (emphasising those i'd noted above as well as some other) as justifying the multiple menstrual needs women have, in the methods they go on to strongly state that “the most commonly reported concerns were not related to management” (line 331).

This this the reality of women's experience in this context, or the measure? Because this doesn't seem consistent with the qualitative results?

Further, I methodologically challenge the inclusion of items that do not discriminate well between experiences. Many items have >90% of women endorsing ‘never’ (in the context where the measure was developed!). Borrowing from item response theory, the ‘difficulty’ of each item here is questionable. It is difficult to see how this measure could be used to assess improvement (even in the context it has been developed) given this ceiling effect when sores are already so high and some items have 97% of respondents endorsing ‘never’ (M47). At a minimum, readers need to be alerted to this issue and it needs to be comprehensively addressed in limitations, but I suggest authors reflect on what these items contribute.

  1. The use of rejected scale items to test validity reveals an inappropriate post-hoc approach to validation.

In a cross-sectional analysis of scale validity I would expect to see an analysis of the relationship between the scale and scale sub-scales and theoretically linked concepts – these might be measures of the same or a very similar concept (convergent validity), different concepts (discriminant validity) or concepts that we would theoretically hypothesize to be associated with the concept as defined.

At places in the results I am not sure which of these the validity tests are meant to be assessing. For example, burning or itching in the vaginal area is going to be very likely if someone has already reported that they have a wound from their cloth/pad (that is probably going to burn). But these aren’t defined as measures of the same concept (convergent validity) but then for a hypothesized association this seems confounded by the similarity of the questions. 

I would expect the concepts for validity testing to be defined a-priori and measured alongside the draft scale items. It is not clear throughout the results which correlates were defined a-priori. At least two were rejected items FROM the measure – clear evidence that these were selected post-hoc. Given the authors would have had to explore the relationship between scale items to develop the set of items, this lends itself to the biased selection of validation items and p-hacking. While I’m sure this would not have been the authors intention, it is still not appropriate to be present ad-hoc analyses of a potential scale item as a validity test. It also raises questions about the clarity of the original construct – if items can then be used as separate concepts against which to test the validity of the scale.

The selection of items for validity testing needs to be more clearly defined in methods, and listed more transparently in the main text not only in supplementary materials so that readers can interrogate these relationships themselves, rather than relying on authors selected results to present.

The removal of post-hoc validity tests with scale items means significant revisions to the discussion are needed. Echoing back to the issue with concept definition – authors discuss at length in the discussion the consequences for women’s work due to menstruation (all of paragraph 3 – line 467-475). This was originally included as a scale item. This again raises issues with concept definition – if this had fit with the factor structure – how would the scale have been used to predict work attendance when this is an item in the scale?? – already there is at least one item in the scale that ‘double dips’ and may be falsely driving some of this association “difficulty working because of wound from cloth/pad”.

Other concerns

The authors provide a very clear description of the sampling and I like the figures in text and supplementary that clarify the flow of the development process and sampling design/relation to the wider study.

Attention to limitations in the discussion

Authors need to significantly expand the limitations section of the discussion. While there are many strengths in the sampling design and multi-step development process, there are many limitations of the measure here. These need to be transparently reported and limitations outlined.

Authors haven’t tested test-re-test reliability, there is little to benchmark against (especially since the data was collected some time ago) in terms of criterion validity [this is just where the field is at - but attention to limitations still needed], there are only cross-sectional analyses reported and it isn’t always clear where hypothesized relationships for validity testing have come from. (Why didn’t authors look at the association with psychological wellbeing – as we know this was measured as an independent concept for the sanitation insecurity study?)

Model fit

For the CFA – the model fit statistics for RMSEA are ‘fair’, not good. Good being <0.05, and fair being >0.05, <0.08. We would expect good fit for TLI and CFI to be >0.95, but for the CFA these are 0.93, dropping from the EFA. More transparent reporting here for readers not quickly familiar with these expectations is needed.

Last 30 days vs last menstrual period?

In methods authors report the tool asked women to report on their experience in the "last 30 days"? However in Appendix A this is "the last 2 menstrual periods"? Which is correct?

Since women self-defined if they were 'currently menstruating' - how were the questions answered if they hadn't menstruated in the last 30 days?

Different menstrual materials

Why did women using disposable pads have lower scores across so many factors? This seems odd, why would concerns about menstrual health and fertility be related to disposable pad use? (socio-economic status/education?)

Were all questions answered by all respondents. 204 (23% of) respondents reported using disposable pads – how did they answer items M21, and M25 which are about washing and drying reusable materials (cloth)? How will future users of this scale be able to incorporate these items in other populations with higher pad usage?

A strength of the paper indeed is the contextual grounding in this context, however this means authors must also address comprehensively in the discussion the limitations of this measure for other settings.

Tension before menstruation

Must is made of ‘tension before menstruation’ in the discussion, but I’m confused about what this means. ‘tension’ elicits ‘pre-menstrual tension’ as in menstrual-cycle caused stress/anxiety/moodiness before and at the start of the menstrual period. Is this how this is understood in this context. If so, I think the paper needs more information on why would we expect this to be associated with menstrual insecurity and the sub-scale scores. If it is understood differently in this context (anxiety about menstruation?) then this needs to be clearly defined for the reader – I’m sure I’m not the only one who will read this as PMS.

Cognitive interviews

Authors should clarify in limitations that “cognitive interviews” were only undertaken with the data collection team, not with actual participants. Cognitive interviews with participants may have elucidated other challenges or changes to questions and may be needed in future use of the scale, particularly were it to be used in a new context. It should be made clearer to readers who the group was for the cognitive interviews, to avoid confusion where one would often expect these to be done with participants. It wouldn’t be unusual for this kind of translation exercise to happen in any data collection training for a measure/survey and it often wouldn’t be framed as a ‘cognitive interviews’ with data collection staff.

Factor scores

The authors use mean scores, not factor scores. The use of ‘factor scores’ throughout is likely to cause confusion as conventionally this is the weighted scores generated from the factor analysis. I do not believe it is appropriate to call mean scores factor scores.

Minor points

Table 4 – the total score is wrong – it starts with 8? I thought the scale was 1-4.

Citation 8 has 76 studies, 87 reports (ie. Some studies were reported across multiple publications) – this should be accounted for in calculating the proportion of studies on adult women.

Author Response

7 April 2020

To Whom It May Concern:

RE: ‘Assessing Women’s Menstruation Concerns and Experiences in Rural India: Development and Validation of a Menstrual Insecurity Measure’ (manuscript: ijerph-774592).

We appreciate the time provided in reviewing our manuscript. Below, please find a point-by-point response to all comments, which proved helpful in improving the quality of this manuscript. We have copied refered text sections into the responses below with line numbers, and have also made all changes in track changes on the manuscript document for reviewing ease.

We hope you find these revisions acceptable. Please do not hesitate to contact me with any questions.

Sincerely,

Bethany A. Caruso 

REVIEWER 1 

Reviewer 1 Overall Comments and Suggestions for Authors

This manuscript reports on the development of a ‘Menstruation Insecurity’ measure which collects data on an array of menstrual-related discomforts and difficulties within a single scale.

I do not believe this can be published in its current form, and significant revisions, including significant updates to the validation analyses would be required for this to be accepted.

I have three most significant concerns (the conceptual clarity of the work and framing of some of the sub-scales, the lack of endorsement of many scale items in the quantitative survey and thus disconnect with the qualitative development work, and the post-hoc selection of rejected scale items for validation), along with a collection of other questions/issues.

I have provided more information on my concerns related to these below.

Author Response: We appreciate the time the reviewer has devoted to this review and are grateful for the suggestions. We are confident that the revisions made based on these suggestions, and the other reviewer, have strengthened the work considerably.

Specific Comments

1.A. Conceptual clarity

1.A.1. Concept definition & fit with research and theory

1.A.1.a. As a reader, the concept of ‘menstruation insecurity’ was not clear. I could not understand the source of the definition, its relationship to other research in menstrual health and hygiene, theory, or the linked sanitation insecurity research.

Authors define menstrual insecurity as “when, as a result of menstruation, social, environmental and biological conditions constrain resources, capacity, confidence, agency, and dignity to care for the self and pursue the activities of daily life.” Notwithstanding that the sentence itself is hard to read, it isn’t clear if ‘menstrual insecurity’ is the social, environmental and biological constraints related to menstruation, or these constraints only when they impact capacity, confidence, agency and dignity (none of these concepts are represented in the scale items, certainly not confidence or agency). Or are these menstrual insecurity only if they also impact caring for the self and pursuing daily activities?

It reads as “any challenge impacting menstrual experience, any part of menstrual experience, and any outcomes of difficulty managing menstruation.” While this may fit with the authors approach of generating items from free listing interviews without imposing an external structure, it does not fit well with a theoretically cogent concept.

It is particularly confusing in the context of the Introduction. Authors do a fantastic job of introducing the need for quantitative measures to further the objectives of menstrual health research, they then go on to contextualize this need in light of measuring menstrual experiences using the framework presented in a recent systematic review of qualitative studies. On line 57 authors suggest that they will “develop a measure to assess menstruation experiences, which can be used independently and to determine how experiences influence health, employment or other impacts” – yet in their definition of menstrual insecurity authors clearly include impacts on everything from confidence and agency to employment and participation in daily life. The definition of menstrual insecurity covers from antecedents of menstrual experience (line 50), experience (line 51) and consequences on health and wellbeing (line 53).

How can the measure be used to assess the association between menstrual insecurity and impacts on health, employment or other impacts, when the concept of menstrual insecurity already includes those impacts? (items in the final scale include having to be separated from one’s usual bed – presumably engagement in normal daily life – an impact – and physical health harms – developing wounds due to cloth). Would it really be appropriate to test the association between menstrual insecurity and impacts if the menstrual insecurity measure already includes some of these impacts? Won’t we be generating false associations by conflating our predictor with our outcome? 

The definition of menstrual insecurity is also a curious fit with the authors definition of sanitation insecurity from 2017. In the same program of work, authors define sanitation insecurity as “insufficient and uncertain access to socio-cultural and social environments that respect and respond to the sanitation needs of individuals, and adequate physical spaces and resources for independently, comfortably, safely, hygienically, and privately urinating, defecating and managing menses with dignity at any time of day or year as needs arise in a manner that prevents fecal contamination of the environment and promotes health”  This definition seems much clearer.  Similar to menstrual insecurity this definition focuses on social and physical contributors, in this case to comfortable and safe sanitation behaviors. Unlike the definition of menstrual insecurity it doesn’t then extend to the consequences. It is more clearly focused on the range of antecedents and anxieties about lacking access to social and physical resources - whereas the definition of menstrual insecurity evokes a range of other concepts.

1.A.1.a. Author response: As all of the above paragraphs relate to the definition, we have aimed to respond to the points raised in the response below.  

  • The reviewer has pointed out some important points regarding ‘Menstrual Insecurity’ as a concept. This concept requires more discussion, and as a first step we have moved the the definition from the introduction to lead off the methods section in order to expand.

  • We have modified the definition. The reviewer is correct in that the definition should not include impacts, but separately explain and hypothesize the association with potential impacts elsewhere. Several of the above paragraphs re-iterate this point, and the point is well taken. The revised definition pulls these concepts out. The paragraph explaining the revised definition is as follows, and is explained to be intentionally broad and preliminary to reflect the ground-up approach and also to enable iteration and refinement through further work.

‘We refer to the compliment of concerns and negative menstruation-related experiences women in this population face as ‘Menstrual Insecurity’. Following best practice [28], we offer a preliminary definition of Menstrual Insecurity to be ‘the suite of social, environmental, and biological concerns and negative experiences resulting from menstruation’. Our intentionally broad, preliminary conceptualization is informed by efforts to create culturally-grounded measures of food[29-31], water[32-34], and sanitation[35] insecurity. Researchers in these fields argue for extending understanding of insecurity constructs beyond biological needs to include the broader socio-cultural context, like social needs and experiences, which may impact well-being [31,32,34,36,37]. Our definition is intentionally broad and described as preliminary as we consider Menstrual Insecurity to be a concept to be iterated upon as research evolves, which has been demonstrated with the concepts of Water Insecurity[38] and Menstrual Hygiene Management[12,23].’ (lines 171-248)

1.A.1.b. Researchers in menstrual health/hygiene can and should debate the merits of more comprehensive vs specific measures of concepts  - “lumping” vs “splitting” and the role of sub-scales vs alternative scales for measuring different concepts and it is likely different research and programmes would call for different approaches. I am somewhat uncomfortable with authors suggesting that this measure is the only thing expanding a focus away from menstrual management - this has been done across a range of comment and qualitative research - that these haven't been included in a single measure but across multiple measures isn't exactly a limitation. Toning this down would be appropriate.

1.A.1.b. Author response:

  1. The reviewer also notes the importance of researchers working in the field of menstruation to be involved in debating the the work and approach. Indeed. Under no circumstances do we expect that the concept we are discussing and presenting will be stagnant. On the contrary, we have aimed to be very upfront about the limitations of this work—expanding the limitations even further even since the reviewer first read the paper. Because of the noted limitations, as well as areas that we explicitly note should be improved, we hope that this concept starts a discussion in the field and continues to evolve and get better over time.

The reviewer specifically raises the issue of ‘lumping’ vs ‘splitting’ concepts. We explicitly note that two factors need further development, but have also added the following indicating that separate scales may be useful, depending on emergent learning from the field:

It may be useful for each of these factors to be developed further as independent scales, depending on context, what may be of greatest utility to researchers and practitioners, and future learning in the field.’ (lines 1012-1015)

As noted in the paper, we have looked to the fields of food and water insecurity for guidance. Concepts of food and water insecurity have changed substantially over time. These concepts, and associated measures, did not emerge in a perfect state, but evolved through research and discourse. They have been carefully and slowly iterated upon. Terrific efforts have been undertaken in multiple sites by myriad researchers to improve upon what has previously been known and understood. The HWISE-RCN (Household Water Insecurity Experiences- Research Coordination Network) is a wonderful example. The researchers engaged in this effort have improved the concept and rigor of water insecurity work. Still, the effort had to start somewhere and grow from something.

We envision our work as a starting point. We hope the definition is debated, and that the overall concept and measure continue to evolve, grow, iterate, and improve. Moreover, we hope that other measures continue to emerge in the field of menstruation research as a whole. This field has had little investment to enable measurement development. We hope that this work inspires effort and investment in further measurement development in the field, like that which has enabled the HWISE-RCN and its members to thrive. Indeed, the initial work undertaken in that field was not perfect, but started a critical line of work.

  1. The reviewer states that they are ‘somewhat uncomfortable with authors suggesting that this measure is the only thing expanding a focus away from menstrual management - this has been done across a range of comment and qualitative research - that these haven't been included in a single measure but across multiple measures isn't exactly a limitation. Toning this down would be appropriate.’

We find this comment confusing, and a bit baffling. The reviewer has not offered a quote or line or paragraph from our paper to show where we have claimed our work to be ‘the only thing expanding a focus away from menstrual management’. Rather, we have simply aimed to describe how our approach is different than other work that focuses on management alone, and we do not feel that the way we have done this is delivered with the tone that the reviewer suggests. It is standard practice to discuss other work, to put work in context, to compare. This is what we have done. What we have written to differentiate our work is as follows:

To fill the need to understand prevalence and frequency of menstruator’s concerns and experiences, we aimed to develop a culturally-grounded measure to assess the suite of concerns and negative menstruation-related experiences that women in rural Odisha voiced. The measure is inclusive of—but not exclusive to—experiences managing menses; a scale assessing menstrual needs among adolescents has been created[27]. This broad approach, inclusive of participant-identified concerns and experiences, will enable understanding of the range of issues women in this population have and can enable future assessments of how concerns and experiences may impact menstruator’s lives and inform program development.’ (lines 161-168)

As we note throughout our response to the reviewer, we have aimed to be very transparent about the limitations of this work and how it can be improved. We would be dismayed to present our work in the tone that the reviewer suggests and would certainly edit our work if it did. If the reviewer wishes to share specific passages that they think are troublesome or require ‘toning down’, we would certainly revisit the text and edit as appropriate.  

1.A.1.c. Authors here argue their approach here is “more inclusive” but don’t clarify the boundaries of their measured concept and the relationship to other concepts in menstrual research (e.g., confidence/self-efficacy, social support). It is stated that menstrual insecurity includes social and physical environment antecedents, which it does include some, but it omits others (likely because these weren’t raised in the free listing interviews in this particular context) – these should be addressed at some point in the paper – there is limited info on social support or environmental infrastructure which we might think of immediately as social and physical antecedents of menstrual experience.

1.A.1.c. Author response: The reviewer takes issue with the term ‘more inclusive’: ‘Authors here argue their approach here is “more inclusive” but don’t clarify the boundaries of their measured concept and the relationship to other concepts in menstrual research’.

Our use of ‘more inclusive’ was intended to reflect the inclusion of participant’s voices in the creation of the measure, not the concept. We have modified that specific sentence for clarity as follows:

This broad approach, inclusive of participant-identified concerns and experiences, will enable understanding of the range of issues women in this population have and can enable future assessments of how concerns and experiences may impact menstruator’s lives and inform program development.’ (lines 165-168)

1.A.1.d. Authors should return to the concept definition in the discussion to further clarify the concept for measurement. It is not straightforward to understand so readers are going to need some help in how the factors and items identified accurately reflect menstrual insecurity. Especially when the definition includes concepts like confidence, agency and dignity, but the measure includes none of these concepts.

 1.A.1.d. Author response: As noted above, we modified the definition in the methods and also revisit the definition in the discussion as follows:

‘We used a rigorous, mixed methods approach to create and validate a culturally-grounded measure of Menstrual Insecurity, which we defined preliminarily as ‘the suite of social, environmental, and biological concerns and negative experiences resulting from menstruation’. The measure was designed to reflect the menstruation-related concerns and negative experiences of women from rural Odisha, India, and to assess the frequency of those concerns and negative experiences among a representative sample. Through our qualitative phase, we identified four themes that we hypothesized would emerge as factors that reflect conditions of women’s social and physical environments, and their individual experiences of menstruation, extending beyond but including management. Five factors emerged that deviate slightly from the four hypothesized, but are theoretically plausible and do reflect the social environment (Restrictions), the social and physical environment collectively (Management), and individual experience (Symptoms; Menstrual Cycle Concerns; Bodily Concerns) as noted in the definition.’ (lines 803-814)

1.A.2. Sub-scale/factor labeling

1.A.2.a.  Related to the concept, I am confused by some of the labels used for the factors/sub-scales. “Menstrual health” has been used much more comprehensively than only to refer to menstrual cycle irregularities (e.g., UNICEF 2019, FSG report). Many now refer to the study of menstrual health and include social and psychological health as part of health (as indicated by the WHO definition of health). Given the context of research in this field I would strongly advise against labeling this factor as ‘menstrual health’ – I think it would cause a lot of confusion. In the paper I wasn’t sure if authors were conceptualizing this factor as really capturing concerns about menstruation due to a lack of menstrual cycle knowledge, or if this factor actually captures women with disordered menstruation. Just as the overarching concept could do with more clarity, so too could the diverse factors included in the very broad scale.

1.A.2.a.  Author Response: We appreciate this point. The concept of ‘Menstrual Health’ has indeed been evolving and we agree that this term/label should be revised. Given the items in this factor, we have re-labeled this ‘Menstrual Cycle Concerns’.

1.A.2.b.  I don’t think it is appropriate for factor 5 to be titled ‘consequences’. This again is likely to lead to misinterpretation that this captures consequences of menstrual experience for women’s lives. It captures one physical consequence of difficulties with materials – wounds, difficulties with work, only in relation to this wound, and worries about odour.

1.A.2.b.  Author Response: We understand the points raised. We have reviewed and renamed this factor ‘Menstruation-related Bodily Concerns’ (which we note to be referred to as ‘Bodily Concerns’ for parsimony after official naming in the paper.

1.A.2.c.  I think more clarity is needed on the 'symptoms' concept. Unlike the 'menstrual health' factor which is framed around concerns about irregularities - the 'symptoms' factor is just about if someone experiences menstrual pains. This seems like more of a biological characteristic. I'm not convinced of the value of including it in this single measure vs separately - but more than that - is this something that is going to respond to interventions? Absolutely we need to pay more attention to women's pain but even an intervention that provided better diagnosis and access to pain relief - would women answer this question differently? They would still 'experience' the onset of pain - it just might be less impacting or distressing because they have better management options. They might experience pain for less time (e.g., they take a pain-killer) but are the questions here going to be sensitive to that?

1.A.2.c.  Author Response: We have expanded our discussion of the ‘Symptoms’ factor. First we note the limitation that the sub-scale includes only two items, and is not comprehensive of all symptoms. We then acknowledge that the items also only assess if women experience symptoms, but not if these are somehow concerning for women. We then indicate how we would make revisions to this sub-scale in future research by not only expanding the items to include more symptoms, but by assessing if the symptoms were concerning or worried women. We also propose the creation of an index to not only assess if women had a range of symptoms, but how intense the symptoms are. Our addition to this section is as follows (and also includes text about the Bodily Concerns Factor):

‘While the Symptoms and Bodily Concerns sub-scales do provide some insight, they also have limitations. The symptoms factor only has two items, limiting its comprehensiveness, and does not enable assessment of whether or not these symptoms are concerning for women, but just the frequency of occurrence. In future work, we would aim to revise this sub-scale by specifically asking women if they were concerned about a more expansive list of symptoms. Further, we think it also would be useful to create a ‘Menstruation Symptoms Index’ to accompany this sub-scale. The index could be used to determine which, of a range of symptoms, women experience, and how intense the symptoms are. Similarly, we would revise items in the Bodily Concerns sub-scale to more explicitly assess worry or challenges related to the body as a result of management, rather than simply asking about an experience (e.g. got wounds from cloth).’ *lines 849-934)

1.B. The poor endorsement of many scale items, particularly the disconnect between the quantitative findings and what was identified through the qualitative process calls into question the validity of the scale and the authors strong claims about a ‘grounded’ and valid measure

1.B.1. The measure was developed through FGDs and free listing interviews. This grounded process is a strength of the study, and prioritizes women’s concerns in this context.

However, the lack of consistency between the concerns that seem to have been raised during the qualitative component, their weighting in exercises during development, and then the endorsement of items in the quantitative scale is highly concerning.

1.B.1. Author Response: We appreciate the reviewer’s comment that the use of FGDs and free listing interviews as part of the grounded process is a strength of the study. In terms of the concern related to the ‘lack of consistency’ between the qualitative and quantitative, we address this point later where it is brought up and further iterated upon. (See Author response 1.B.3).

1.B.2. It would be helpful for readers to include in methods (ie. Lines 168-171) the thematic groupings of concerns as identified through the FGDs and FLIs – this could then be compared to the emergent factors. I don’t believe referral to supplementary materials in another paper is sufficient for the readership of this paper. How did the emergent themes from the qualitative study inform the concept of menstrual insecurity and the hypothesized factors for the scale, and the items for inclusion? Presumably the authors had some a-priori concepts they had developed and identified if these would be appropriate for measurement within a scale of menstrual insecurity (i.e., which concerns raised fit with the 'menstrual insecurity' concept). In results authors discuss balancing model fit with ‘theoretical fit’ of the data. From the 2017 paper it looks like some of these groups were ‘bathing’ ‘washing cloth’ ‘drying cloth’ – yet none of these appear as factors (they do appear as items).

This information could be provided in preference to other chunks of text that aren’t that useful for readers of this study. For example lines 113-121 could be significantly shortened – while it makes sense to contextualize this work in the context of others with the same program of work this is confusing for the reader and not essential information for this paper which already requires a lot of attention from the reader without further distractions. I’m not sure you need all the life-stage groupings in Table 1 – especially since there were few differences in the scale across – I think this would be more appropriate for supplementary materials and you could then have more scale relevant information in the main text.

1.B.2. Author Response

a). Additional information in the methods. We have added additional information in the methods to included thematic groupings as suggested. Further, we have now provided a table in the supplement that includes the initial groupings of the survey items by identified themes, as well as the result post EFA/CFA. The specific methods information appears in ‘2.4.2 Phase 1, Stage 2: Item Identification’ and is noted below:

‘All FLIs were analyzed first to identify the full suite of concerns or negative experiences participants believed women in their community to have. We then determined the proportion of participants who reported each concern or experience. Thirty-two unique menstruation concerns were shared by 67 (97%) of the 69 women interviewed. FGDs supported information gathered in FLIs (See Caruso 2017, including supplemental tables, for list and frequencies of concerns)[22]. Next, concerns were analyzed thematically to identify emergent grouping. We identified and sorted concerns into four themes, which included: Management, Restrictions or challenges to normal activities, Social needs and constraints, and Well-Being. Wording of the identified concerns in each theme was then adapted to create draft items. (See Supplement Table 1 for Survey Items by theme) The initial list of items included the full range of concerns noted, however monsoon-specific items were omitted due to irrelevance during the following anticipated data collection period. We hypothesized that these themes would emerge as factors in the final measure.’ (lines 317-340)

b). We disagree with the reviewer suggestion to remove lines 113-121. Given what is said about those lines, and what we have written, we are confused by the suggestion and think it is possible the reviewer added the wrong line numbers. As such, no changes have been made directly to those lines.

We also disagree with the suggestion to remove table 1 and add it to the supplement. Among those working in the field of menstruation and in gender more broadly, there is a recognized need to not consider women to all be the ‘same’ when they are not. Further, our sampling was intentional and aimed to get representation of the groups identified. We do not think it would then be appropriate to present the data in a uniform fashion. Doing so would further perpetuate the notion that all women are the same.  A table without disaggregation would take up equivalent space. Finally, we have added quite a lot of scale-relevant information in the text as is and we are unclear what the reviewer thinks is missing or needs to be added in its place.

1.B.3. From the 2017 paper, and the introduction of this paper it looks like women “reported 32 unique menstruation concerns". The most endorsed concerns were concerns about bathing (52%), washing (51%) and drying cloths (46%). And yet, in the quantitative survey and presentation of scale results, 82% of women reported never having trouble finding a place to wash their cloth, 77% reported no difficulties bathing, and 85% reported no issues finding a place to dry their cloth. This is never addressed in this paper or in the limitations.

I find it deeply concerning that the authors overlook this disparity between the qualitative findings and scale results. If this measure is truly grounded in the experiences of women in the study context, then how can the scale have such low endorsement of many items? In the introduction authors report the concerns raised in the qualitative (emphasising those i'd noted above as well as some other) as justifying the multiple menstrual needs women have, in the methods they go on to strongly state that “the most commonly reported concerns were not related to management” (line 331).

This this the reality of women's experience in this context, or the measure? Because this doesn't seem consistent with the qualitative results?

1.B.3.  Author Response: The reviewer indicates that there is a disconnect or lack of consistency between the qualitative and quantitative findings. Upon a re-read of the introduction section that pulls from the qualitative work, we understand how the reviewer has this understanding. In our qualitative work, we describe  in the methods that ‘We asked women to list concerns ‘women in this community’ had when urinating, defecating and menstruating, and probed to identify temporal influence (eg, diurnal, seasonal) and variation across pregnancy and dependency status.’

We intentionally asked women of their perceptions of experiences and concerns women in their community had, not just to identify their own. This approach was taken because of taboos associated with discussing menstruation, urination, and defecation, and our assumption that women would feel more open about speaking of experiences and concerns if they did not feel compelled to speak of their own personal experiences. As such, the concerns reported in the qualitative work reflect broad perceptions of women’s concerns and negative experiences in the community. In other words, that 52% of our qualitative participants note concerns about bathing at menstrual onset does not mean that 52% of those participants each individually had that concern. Rather, it means that 52% of the respondents indicated that they perceived bathing at menstrual onset to be a concern for women in their community. As a reminder, a primary purpose of this work was to understand the prevalence and frequency of the issues noted qualitatively at an individual level.

We have clarified language in the introduction for clarity as follows:

‘Women participating in qualitative interviews in rural Odisha, where the present research took place, reported 32 unique menstruation concerns that they believed women in their community experienced, including concerns about…’ (lines 152-154)

Further, we have revisited this difference between the qualitative and quantitative in the discussion (See response to next comment).

1.B.4. Further, I methodologically challenge the inclusion of items that do not discriminate well between experiences. Many items have >90% of women endorsing ‘never’ (in the context where the measure was developed!). Borrowing from item response theory, the ‘difficulty’ of each item here is questionable. It is difficult to see how this measure could be used to assess improvement (even in the context it has been developed) given this ceiling effect when sores are already so high and some items have 97% of respondents endorsing ‘never’ (M47). At a minimum, readers need to be alerted to this issue and it needs to be comprehensively addressed in limitations, but I suggest authors reflect on what these items contribute.

 1.B.4.  Author Response:

As the reviewer has suggested, we have added to the discussion to highlight the high proportion of women indicating ‘never’. We think it important to re-emphasize here, and in our paper, that the qualitative work queried participants about their perceptions of women in the community, whereas our survey asked women about themselves individually. We have added the following:

‘All the retained items in the Menstrual Cycle Concerns sub-scale had over 90% of women respond that they ‘never’ had the concern or experience. Comparatively, 27% of qualitative participants perceived irregular menstruation to be a concern for women in general [22]. While a high proportion of ‘never’ responses for these and other items may seem at odds with our qualitative work, which showed higher endorsement, differences were expected. The current measure assesses concerns at the individual-level, whereas the qualitative work asked about perceived concerns among women in the community in general, a deliberate approach used to capture a range of perceived concerns and to enable participants to feel more comfortable talking about challenging topics. Our quantitative assessment demonstrates that, though concerns or experiences may be widely recognized qualitatively in a community, many are not or are infrequently experienced at an individual level, further highlighting the value of mixed methods assessments.’ (lines 983-993)    

Related to the reviewer’s comment about assessing improvement, yes, there is a ceiling effect, but that is also useful for programming. For example, what we see here is that, despite a general perception among women engaged in the qualitative work that irregularity is a concern for women in their community (27%), we learn that a low proportion of women report these concerns when queried individually. As such, programs can use data generated (at a baseline) to inform programmatic directions and priorities. It would make more sense, obviously, to invest in areas that have potential for improvement.

1.C. The use of rejected scale items to test validity reveals an inappropriate post-hoc approach to validation.

In a cross-sectional analysis of scale validity I would expect to see an analysis of the relationship between the scale and scale sub-scales and theoretically linked concepts – these might be measures of the same or a very similar concept (convergent validity), different concepts (discriminant validity) or concepts that we would theoretically hypothesize to be associated with the concept as defined.

At places in the results I am not sure which of these the validity tests are meant to be assessing. For example, burning or itching in the vaginal area is going to be very likely if someone has already reported that they have a wound from their cloth/pad (that is probably going to burn). But these aren’t defined as measures of the same concept (convergent validity) but then for a hypothesized association this seems confounded by the similarity of the questions. 

I would expect the concepts for validity testing to be defined a-priori and measured alongside the draft scale items. It is not clear throughout the results which correlates were defined a-priori. At least two were rejected items FROM the measure – clear evidence that these were selected post-hoc. Given the authors would have had to explore the relationship between scale items to develop the set of items, this lends itself to the biased selection of validation items and p-hacking. While I’m sure this would not have been the authors intention, it is still not appropriate to be present ad-hoc analyses of a potential scale item as a validity test. It also raises questions about the clarity of the original construct – if items can then be used as separate concepts against which to test the validity of the scale.

The selection of items for validity testing needs to be more clearly defined in methods, and listed more transparently in the main text not only in supplementary materials so that readers can interrogate these relationships themselves, rather than relying on authors selected results to present.

The removal of post-hoc validity tests with scale items means significant revisions to the discussion are needed. Echoing back to the issue with concept definition – authors discuss at length in the discussion the consequences for women’s work due to menstruation (all of paragraph 3 – line 467-475). This was originally included as a scale item. This again raises issues with concept definition – if this had fit with the factor structure – how would the scale have been used to predict work attendance when this is an item in the scale?? – already there is at least one item in the scale that ‘double dips’ and may be falsely driving some of this association “difficulty working because of wound from cloth/pad”.

 1.C.1. Author Response:

  1. Concepts for validity tests. We do present analyses of the relationship between the sub-scales and theoretically linked concepts. The specifics of these analyses were previously expanded upon in text in the supplement. We have moved text back to the methods section where validity is discussed (section 2.6.4). We believe this inclusion will add clarity to the results as well.

Types of validity tests. The methods section is also more clear regarding the types of validity test conducted.

The expanded section in the methods is as follows:

‘As previously noted, content validity was assessed by two external experts and the two Research Assistants (RAs) who led Free-list interviews and FGDs, and face validity was assessed by both the RAs and the enumerator team. Convergent validity was determined by assessing correlations between each subscale score and items from the survey that asked about thematically similar concepts. Specifically, for Subscale 1 (Restrictions), we assessed correlation of scores with traveling alone; Subscale 2 (Management) scores with having resources to change menstruation material; Subscale 3 (Symptoms) scores with infection symptoms, and Subscale 4 (Menstrual Cycle Concerns) scores with experiencing regular menstruation. Additionally, known-groups validation was used to assess discriminant validity, with two sample t-tests being used to verify whether Subscale 5 (Menstruation-Related Bodily Concerns) scores differed by those reporting burning or itching in the vaginal area.’ (lines 486-496)

  1. The reviewer notes: “While I’m sure this would not have been the authors intention, it is still not appropriate to be present ad-hoc analyses of a potential scale item as a validity test.”

To be clear, we do not present the the ad-hoc analyses as a validity test. The methods for the validity tests are explicitly noted in the methods section about validity.  We discuss presenting these final scores disaggregated by these item scores in the section on scoring.  However, we have revised that section to be more clear that these were rejected scale items and the decision to use them was not determined a-priori:

We also assessed differences in mean scores by responses to two items omitted from the scale. One assesses tension at menstrual onset, meaning a general feeling of anxiety about menstruation (not pre-menstrual stress or PMS, which can occur one to two weeks prior to menstruation), and the other assesses perceived difficulty doing regular work during menstruation. Analyses with these two items were not conceptualized a priori but were added after these items were omitted as a preliminary investigation of how these scores may vary by reported experiences of tension and work. ‘ (lines 504-532)

Further, we again re-iterate in the discussion section that these were post-hoc analyses, and further research is needed to assess impacts of menstruation-related experiences on work and well-being. The final line of the first paragraph in the discussion includes the following:

‘Post-hoc analyses demonstrated that Menstrual Insecurity scores were highest among those reporting tension at menstrual onset or difficulty doing regular work during menstruation, and suggest the need for further research.’ (lines 816-818)

We disagree that we need to remove the presentation of the scores simply because they were not identified a priori. We are transparent about limitations and the need for further analyses. We have altered discussion about these concepts in the discussion section.

1.D. Other concerns

The authors provide a very clear description of the sampling and I like the figures in text and supplementary that clarify the flow of the development process and sampling design/relation to the wider study.

1.D. Author Response: We appreciate this positive feedback. We have created these figures to enable clarity about the sampling and development process and are encouraged to hear these have been effective.

1.E. Attention to limitations in the discussion

Authors need to significantly expand the limitations section of the discussion. While there are many strengths in the sampling design and multi-step development process, there are many limitations of the measure here. These need to be transparently reported and limitations outlined.

Authors haven’t tested test-re-test reliability, there is little to benchmark against (especially since the data was collected some time ago) in terms of criterion validity [this is just where the field is at - but attention to limitations still needed], there are only cross-sectional analyses reported and it isn’t always clear where hypothesized relationships for validity testing have come from. (Why didn’t authors look at the association with psychological wellbeing – as we know this was measured as an independent concept for the sanitation insecurity study?)

1.E. Author Response: We have aimed to be expansive in our limitation section, and we have expanded the section further to incorporate these, as well as a few additional points. Specifically, we have added a statement about not being able to do test-retest or assess criterion validity. In addition, we have expanded the section on validity testing; information was previously in the supplement and we have moved back to the manuscript for further clarity.  

The revised limitation section is pasted below:

‘A strength of this research is the ground-up approach. Items emerged from qualitative work with rural Indian women of varied life stages, allowing their voices to shape the measure, and leading to a scale with factors that represent their concerns. We would have liked to have conducted cognitive interviews with women to enable additional feedback, however resources were constrained and we were unable to do so. However, we did carry out a careful review process with the enumerator team members, who are from similar villages as the participants, which provided some additional insights regarding translation. The ground-up approach is also a limitation. Some items may not be relevant in other contexts, and the concerns of women in other settings may not be represented. However, the factors, and many items, align with menstruators’ experiences across settings. Testing the scale, or even specific factors, in other settings and with other populations is needed. Additionally, our scale identified factors important to the menstruation experience beyond management. Two factors, Restrictions and Symptoms, have only two items and future work should develop these factors. It may be useful for each of these factors to be developed further as independent scales, depending on context, what may be of greatest utility to researchers and practitioners, and future learning in the field.

Due to constrained time and resources, we were not able to return to a subsample of participants to re-administer the survey and assess test-re-test reliability, nor were we able to carry out data collection at varied time points to assess predictive validity. Given that there are no comparable measures, we also were not able to assess criterion validity.’ (lines 1001-1019)

As noted by the reviewer, for work on sanitation insecurity, we looked at the association with psychological wellbeing. However, that work was in a separate paper than the paper that described the development of the sanitation insecurity measure. Similarly, for this paper our aim is to thoroughly describe our approach to creating the measure. We plan on looking at the association with wellbeing in a different paper.

1.F. Model fit

For the CFA – the model fit statistics for RMSEA are ‘fair’, not good. Good being <0.05, and fair being >0.05, <0.08. We would expect good fit for TLI and CFI to be >0.95, but for the CFA these are 0.93, dropping from the EFA. More transparent reporting here for readers not quickly familiar with these expectations is needed.

1.F.1. Author Response: We did not explicitly indicate that the CFA model fit statistics were ‘good’ as the reviewer suggests. We did write that the EFA model fit statistics were good, which align with what the reviewer indicates to be good.

In the original document, we wrote the following for EFA, which we think is appropriate:

‘The model produced strong, positive loadings and had good model fit statistics (RMSEA = 0.027; CFI = 0.994; TLI=0.989; Table 3).’ (lines 602-603)

And in the original document, we wrote the following for CFA, which we think is appropriate:

‘All items loaded in the CFA in similar ranges as for the EFA and were significant (Table 3). No additional items were dropped. The model fit statistics provided satisfactory evidence that the factor structure was appropriate for the data (RMSEA = 0.058; CFI = 0.937; TLI = 0.925).’ (lines 658-660)

1.G. Last 30 days vs last menstrual period?

In methods authors report the tool asked women to report on their experience in the "last 30 days"? However in Appendix A this is "the last 2 menstrual periods"? Which is correct?

1.G.1. Author Response: This was mislabeled in the methods section and is correct in the appendix. We have changed the methods to read as follows:

‘The final tool included 40 items that asked women how often they had a particular menstruation-related experience during their last two menstrual periods: never, sometimes, often, or always (See Appendix A for tool).’ (lines 359-361)

Since women self-defined if they were 'currently menstruating' - how were the questions answered if they hadn't menstruated in the last 30 days?

1.G.2. Author Response: As noted above, we had a typo in the methods regarding the previous 30 days. We had women respond based on their experiences during the previous two menstrual periods. We indicated that they could consider the previous two menstrual periods because we anticipated that those who were irregular may not have had a period within a set time period.

1.H. Different menstrual materials

1.H.1. Why did women using disposable pads have lower scores across so many factors? This seems odd, why would concerns about menstrual health and fertility be related to disposable pad use? (socio-economic status/education?)

1.H.1. Author Response: Women using disposable pads had lower scores across three of the five factors. These associations seem pretty obvious given the content of the sub-scales. One factor was ‘Menstruation Management’, which queries about various concerns related to washing and drying materials. Women using pads should not have these concerns, so it absolutely makes sense for their scores to be lower. A second factor is ‘Bodily Concerns’, which asks about irritation from materials and concern about the body smelling. Qualitatively, women discussed how cloth was more cumbersome and bothersome and could hold smell, so this association is quite clear as well.

The third factor, formally ‘Menstral Health’ and now renamed ‘Menstrual Cycle Concerns’. In the discussion, we point out that this sub-factor highlights the need for better information about menstruation. It is likely that women using disposable pads have more access to resources overall. The following has been added to the discussion section:

Women who have access to disposable pads had significantly lower scores for the Menstrual Cycle Concerns sub-scale. It may be that those with access to disposable products may also have better access to other resources, like health care or information (booklets with menstruation information are often provided in packages of commercial menstrual products) and thus do not have as many unanswered questions or concerns as those not using pads regularly.’ (lines 973-978)

1.H.2. Were all questions answered by all respondents. 204 (23% of) respondents reported using disposable pads – how did they answer items M21, and M25 which are about washing and drying reusable materials (cloth)? How will future users of this scale be able to incorporate these items in other populations with higher pad usage?

1.H.2. Author Response: Yes. All questions were answered by all respondents regardless of material type. If these were not concerns these women had in the previous 30 days, they could answer ‘never’. Future users should consider if these are appropriate for their populations, just as they should consider if all items are appropriate for their populations. We re very clear throughout that this work is grounded in a specific population, also discussing this as a limitation. And we note that testing in other populations would be needed.

1.H.3. A strength of the paper indeed is the contextual grounding in this context, however this means authors must also address comprehensively in the discussion the limitations of this measure for other settings.

1.H.3. Author Response: We are certainly aware of the limitation that this ground-up approach brings. We do not expect this to be the final and definitive measure on this topic by any means. That would not be appropriate. Rather, we hope this to be the start of a discourse and for future research to improve on what we were able to do given the resources we had to do it. As such, we have noted several limitations of this approach as follows:

‘The ground-up approach is also a limitation. Some items may not be relevant in other contexts, and the concerns of women in other settings may not be represented. However, the factors, and many items, align with menstruators’ experiences across settings. Testing the scale, or even specific factors, in other settings and with other populations is needed.’ (lines 1007-1010)

1.I. Tension before menstruation

Must [sic] is made of ‘tension before menstruation’ in the discussion, but I’m confused about what this means. ‘tension’ elicits ‘pre-menstrual tension’ as in menstrual-cycle caused stress/anxiety/moodiness before and at the start of the menstrual period. Is this how this is understood in this context. If so, I think the paper needs more information on why would we expect this to be associated with menstrual insecurity and the sub-scale scores. If it is understood differently in this context (anxiety about menstruation?) then this needs to be clearly defined for the reader – I’m sure I’m not the only one who will read this as PMS.

1.I. Author Response: We do not mean for this to be understood as pre-menstrual stress (PMS). The item specifically queries ‘tension at menstrual onset’. Pre-menstrual stress (PMS) occurs one to two weeks prior to menstruation, and is linked to hormones. This specific question, about tension at menstrual onset is in relation to the experience of anxiety about menstruation. Anxiety and stress about menstruation, because of the experiences of and concerns about menstruation, have been widely reported in the qualitative literature and in the noted systematic review. In this context, I think it is already quite clear why tension at menstrual onset is associated with the sub-scale scores.

We certainly do not want this to be confused with PMS, which we understand may be most likely among non-menstruators. As such, clarified the methods section as follows:

We also assessed differences in mean scores by responses to two items omitted from the scale. One assesses tension at menstrual onset, meaning a general feeling of anxiety about menstruation (not pre-menstrual stress or PMS, which can occur one to two weeks prior to menstruation), and the other assesses perceived difficulty doing regular work during menstruation.’ (lines 504-530)

1.J. Cognitive interviews

Authors should clarify in limitations that “cognitive interviews” were only undertaken with the data collection team, not with actual participants. Cognitive interviews with participants may have elucidated other challenges or changes to questions and may be needed in future use of the scale, particularly were it to be used in a new context. It should be made clearer to readers who the group was for the cognitive interviews, to avoid confusion where one would often expect these to be done with participants. It wouldn’t be unusual for this kind of translation exercise to happen in any data collection training for a measure/survey and it often wouldn’t be framed as a ‘cognitive interviews’ with data collection staff.

1.J. Author Response: We noted that items were reviewed with the team using a ‘cognitive interview methodology’ and provide explanation to describe what specifically was done. We did not indicate that we were conducting cognitive interviews. We have updated the description and slightly changed the phrasing to be more clear, which can be found below.

‘Third, as a preliminary means of pre-testing, each item was reviewed with the nine female enumerators hired to administer the survey using a cognitive interview approach [28,47]. In addition to having experience with survey data collection, including studies specific to sanitation and hygiene, all enumerators were from villages similar to those to be engaged in the next phase and were able to comment on the items regarding item content and translation. For each item, the RAs asked a team member to explain what the question was asking in their own words. Other team members were welcome to comment or offer an alternative understanding. The group then discussed, and changes to the item translation were made as needed.’ (lines 348-355)

Additionally, we have added information in the ‘limitations section:

‘We would have liked to have conducted cognitive interviews with women to enable additional feedback, however resources were constrained and we were unable to do so. However, we did carry out a careful review process with the enumerator team members, who are from similar villages as the participants, which provided some additional insights regarding translation.’ (lines 1003-1007)

1.K. Factor scores

The authors use mean scores, not factor scores. The use of ‘factor scores’ throughout is likely to cause confusion as conventionally this is the weighted scores generated from the factor analysis. I do not believe it is appropriate to call mean scores factor scores.

1.K. Author Response: ‘Factor scores’ has been replaced with the term ‘sub-scale scores’ throughout as relevant.

1.L. Minor points

1.L.1. Table 4 – the total score is wrong – it starts with 8? I thought the scale was 1-4.

1.L.1. Author Response: This was a typo and the number has been fixed. It is 1.46 (0.34).

1.L.2. Citation 8 has 76 studies, 87 reports (ie. Some studies were reported across multiple publications) – this should be accounted for in calculating the proportion of studies on adult women.

1.L.2. Author Response: The sentence has been modified for clarity as follows:

‘Only 18% (16 of 87) of the publications included in the review focused exclusively on women’ (lines 143-144)

Reviewer 2 Report

Originality

This study developed and validated a multidimensional measure on menstruation insecurity among women in rural Odisha, India. This scale is the first of its kind and will be useful in assessing women’s concerns and experiences associated with menstruation, identify health inequities, and the impact of public health interventions.

Significance

The results of this research correspond with the application of best practices for the development and validation of scales. I consider the scale to be well validated, reliable and has programmatic implications. However, the authors need to justify whether they are using the scale in its multidimensional form, unidimensional form or both.

Quality of presentation:

The article is written and presented in an appropriate way requiring fewer edits and restructuring. Primarily, the abstract should be written such that the authors account for the objectives, methods, results, findings and conclusions/recommendations of the study. Currently, there is limited information on the methods and results. By reorganizing the abstract, the authors can add up some information about the tests of validity and reliability. The current findings are based on a comparison between known groups, which is just a single aspect of validity. Such results while important might be best reserved for the discussion. The conclusion or recommendation in the abstract is not strong enough to validate the importance of the scale developed. I have provided an example that may seem to be a bit more compelling. “This validated tool is useful for measuring menstruation insecurity, assessing health inequities and the correlates of menstruation insecurity, and targeting programs to improving women’s hygiene”.

Scientific Soundness

  1. In content validation, the authors indicate they engaged RAs in cognitive interviews. It is not clear whether the RA’s were part of the target population. Best practice will require the cognitive interviews to be done with the target population. A subset of the participants should be able to speak to the mental processes associated with their understanding of each of the questions. In this way, the authors will be able to assess the appropriateness of the questions asked.

  1. The authors indicated that changes to the items and translation were made as needed. It is not clear what prompted those changes and what kind of changes were made. This should be clarified.

  1. A review of the validation techniques suggests the author used about four aspects of construct validity but this is not made visible in the paper. At a cursory study, it will appear that the authors did not assess external validity (Predictive, convergent or discriminant). However, a careful read shows these tests were done. Differentiating between the different forms of validity will enhance the logical progression of the paper. Please refer to:

Messick, S. (1995). "Standards of validity and the validity of standards in performance assessment". Educational Measurement: Issues and Practice14 (4): 5–8. doi:10.1111/j.1745-3992.1995.tb00881.x

  1. The authors do indicate that the amount of variation explained by the 19 items is above the 60% threshold. It is not clear what threshold was used for this assessment. The authors should add a citation to back this position. To my knowledge, 70% is the threshold used in most psychometric evaluations. I should indicate that the current variation of 69% is as good as 70%.

  1. Due to the multidimensionality of the scale items, it is possible that all the subdomains could form a unidimensional scale. i.e. all 19 items could be assessed to be unidimensional or form an overarching measure. Hierarchical confirmatory factor analysis is necessary to determine whether this is the case. Also, if the authors intend to use each of the subscales by themselves, then they need to assess whether each of the subscales is unidimensional in the second data. Further, it is possible that factor 3 may be contributing noise to the entire model. The authors may need to test for this using a bifactor model to be sure this is not the case.
  2. The test of reliability makes one to wonder whether the authors tested for the unidimensionality of the entire 19-item scale. If that was not done, it is difficult to understand why they assessed the reliability of the 19 items together. Emphasis should be laid on the reliability of the sub scales or factors. Again, if this test was not done, the authors cannot use the total score of all 19 items as a menstruation insecurity score. Consequently, the need to do a higher order confirmatory factor analysis and test for the unidimensionality of the scale items.

  1. The diagrammer in Mplus produces one of the best figures associated with CFA. The authors should consider providing a diagram output for clarity of the results.

  1. The authors assert that the mean overall Menstruation Insecurity score was 1.46; the authors should clarify whether this is coming from an average of all 19 items. If so, it needs to be justified as confirmatory factor analysis was not conducted directly on all 19 items but sub-factors.

  1. For the construct validity, the authors should consider using a regression model instead of correlations. At minimum, a bivariate regression model. This will also provide the amount of variance explained by each construct in the outcome variable.

  1. Again, the total score as indicated in Table 4 is not possible if the authors did not test for the unidimensionality of all the 19 items. If no new models are added, the authors may have to revise the discussion to show the value of the menstruation insecurity scale based on the sub-scales.

  1. The authors may have to explain in the domain identification how they arrived theoretically at the five domains. Even if it is reported elsewhere, it is important that this is made clear in the paper.

  1. While I appreciate the value of a menstruation insecurity scale, menstruation is a biological phenomenon and not a latent construct. To measure it as a latent construct, the concept needs to be used in relation to a qualifier or context. Hence, I will ask that the authors consider using menstruation hygiene insecurity instead of menstruation insecurity. Alternatively, the authors should clarify the conceptualization of menstruation insecurity.

  1. The authors in the third paragraph of the discussion indicate that “our work shows that menstruation can impact women’s ability to work, an understudied area of menstruation research”. However, this is based on the comparison between known groups, which did not adjust for any control variables. The relationship between the variables could be spurious. Hence, it is best to showcase the significant differences and not draw definite conclusions.

Interest to the Readers

This paper has significant relevance to researchers in the area of water, sanitation and hygiene and has a lot of relevance in measuring health inequities for women in resource poor settings. Due to its novelty, it is very likely that academics, public health practitioners, community health workers, and policy makers will find the tool useful in promoting the health of women of all ages, especially, those that have reached menarche. I have no doubt that this paper will attract a wider readership.

Overall Merit

This study makes an important contribution to research in the area of water, sanitation and hygiene, women's health, and produces a unique tool which can used to assess health inequities, challenges of women in relation to menstruation, and the correlates of menstruation insecurity.

English level

The paper is well written and logically coherent in structure. However, I will ask that the authors are consistent in their use of menarche or period. You may choose to stay with a single description for menstruation. Please pay attention to subject verb agreements e.g. Line 107, line 221

Author Response

27 April 2020

To Whom It May Concern:

RE: ‘Assessing Women’s Menstruation Concerns and Experiences in Rural India: Development and Validation of a Menstrual Insecurity Measure’ (manuscript: ijerph-774592).

We appreciate the time provided in reviewing our manuscript. Below, please find a point-by-point response to all comments, which proved helpful in improving the quality of this manuscript. We have copied referred text sections into the responses below with line numbers, and have also made all changes in track changes on the manuscript document for reviewing ease.

We hope you find these revisions acceptable. Please do not hesitate to contact me with any questions. Thank you for your consideration of this manuscript.

Sincerely,

Bethany A. Caruso 

REVIEWER 2

Reviewer 2 Overall Comments and Suggestions for Authors

2.A Originality

This study developed and validated a multidimensional measure on menstruation insecurity among women in rural Odisha, India. This scale is the first of its kind and will be useful in assessing women’s concerns and experiences associated with menstruation, identify health inequities, and the impact of public health interventions.

2.A. Author Response: We appreciate this positive and encouraging feedback. We are very grateful for the time and careful review provided. We feel the comments provided have strengthened this manuscript very much.

2.B. Significance

The results of this research correspond with the application of best practices for the development and validation of scales. I consider the scale to be well validated, reliable and has programmatic implications. However, the authors need to justify whether they are using the scale in its multidimensional form, unidimensional form or both.

2.B. Author Response: Thank you for this feedback regarding our process. We provided more details below regarding the multidimensional and unidimensional use of the scale (See responses to questions that follow).

2.C. Quality of presentation:

The article is written and presented in an appropriate way requiring fewer edits and restructuring. (a)Primarily, the abstract should be written such that the authors account for the objectives, methods, results, findings and conclusions/recommendations of the study. Currently, there is limited information on the methods and results. By reorganizing the abstract, the authors can add up some information about the tests of validity and reliability. (b)The current findings are based on a comparison between known groups, which is just a single aspect of validity. Such results while important might be best reserved for the discussion. (c) The conclusion or recommendation in the abstract is not strong enough to validate the importance of the scale developed. I have provided an example that may seem to be a bit more compelling. “This validated tool is useful for measuring menstruation insecurity, assessing health inequities and the correlates of menstruation insecurity, and targeting programs to improving women’s hygiene”.

2.C. Author Response: We are encouraged that the reviewer finds the overall presentation of the work to be strong and we appreciate the recommended adjustments. We have outlined below our changes point by point:

  1. Abstract structure. Thanks for this recommendation. The journal has specific requirements to have a maximum of 200 words in a single paragraph that includes background, methods, results, conclusion (all unlabeled). We now see that the abstract is lacking information on methods, and we have added additional information. The new abstract is pasted below these comments for easy reference.
  2. Suggestion to put results in discussion: We are unclear if the reviewer is referring to the main text or abstract here. If main text, we feel it vital to not introduce findings in the discussion and we have elected to keep these comparisons in the results section. If in the abstract, the IJERPH formatting is a single paragraph (no headings/sections) so will simply come after main findings/before final sentence.
  3. Abstract conclusion. Many thanks for this recommendation for stronger wording. We have included a slight adaptation of the recommendation to fit with word count.

‘Qualitative research has documented challenges faced by menstruators, particularly in water and sanitation poor environments, but quantitative assessment is limited. We created and validated a culturally-grounded measure of Menstrual Insecurity to assess women’s menstruation-related concerns and negative experiences. With cross-sectional data from 878 menstruating women in rural Odisha, India, we carried out Exploratory (EFA) and Confirmatory (CFA) Factor Analyses to reduce a 40-item pool and identify and confirm the scale factor structure. A 19-item, five factor model best fit the data (EFA: root mean square error of approximation (RMSEA)= 0.027; comparative fit index (CFI)=0. 0.994; Tucker-Lewis index (TLI)= 0.989; CFA: RMSEA = 0.058; CFI = 0.937; TLI = 0.925). Sub-scales included: Management, Menstrual Cycle Concerns, Symptoms, Restrictions, and Menstruation-Related Bodily Concerns. Those without access to a functional latrine, enclosed bathing space, water source within their compound, or who used reusable cloth had significantly higher overall Menstrual Insecurity scores (greater insecurity) than those with these facilities or using disposable pads. Women reporting experiencing tension at menstrual onset or difficulty doing work had significantly higher Menstrual Insecurity scores.  This validated tool is useful for measuring Menstrual Insecurity, assessing health inequities and correlates of Menstrual Insecurity, and informing program design.’

2.D. Scientific Soundness

2.D.1. In content validation, the authors indicate they engaged RAs in cognitive interviews. It is not clear whether the RA’s were part of the target population. Best practice will require the cognitive interviews to be done with the target population. A subset of the participants should be able to speak to the mental processes associated with their understanding of each of the questions. In this way, the authors will be able to assess the appropriateness of the questions asked.

 2.D.1. Author Response: We appreciate this comment as it clarified that our description of this activity was not clear. The RAs (Research Assistants) conducted the qualitative data collection, and they did review the items to be sure that the full suite of concerns they heard were included.

As a next step, the RAs reviewed the survey with the nine women who were hired to carry out the survey data collection (the enumerator team), using a cognitive interview methodology. The nine enumerator team members were all from rural villages in Odisha near and similar to the villages engaged in the study.

We have aimed to clarify this in the manuscript and the section now reads as follows:

‘Second, the two research assistants who conducted the FLIs and FGDs provided additional assessment of content validity, specifically face validity, by confirming that items captured the range of concerns they heard and that wording was appropriate. The two RAs independently translated the items, then compared the translations item-by-item, and discussed alternative phrasings before finalizing translation. Third, as a preliminary means of pre-testing, each item was reviewed with the nine female enumerators hired to administer the survey using a cognitive interview approach [28,47]. In addition to having experience with survey data collection, including studies specific to sanitation and hygiene, all enumerators were from villages similar to those to be engaged in the next phase and were able to comment on the items regarding item content and translation. For each item, the RAs asked a team member to explain what the question was asking in their own words. Other team members were welcome to comment or offer an alternative understanding. The group then discussed, and changes to the item translation were made as needed.’ (lines 344-355)

Additionally, we have added information in the ‘limitations section:

‘We would have liked to have conducted cognitive interviews with women to enable additional feedback, however resources were constrained and we were unable to do so. However, we did carry out a careful review process with the enumerator team members, who are from similar villages as the participants, which provided some additional insights regarding translation.’ (lines 1003-1007)

2.D.2. The authors indicated that changes to the items and translation were made as needed. It is not clear what prompted those changes and what kind of changes were made. This should be clarified.

 2.D.2. Author Response: There was a minor typo in the sentence noted. Changes were not made to the item and translation, but to the item translation. The items themselves were not changed.

It has been changed to read: ‘The group then discussed, and changes to the item translation were made as needed’. (Section 2.3.3.)

2.D.3. A review of the validation techniques suggests the author used about four aspects of construct validity but this is not made visible in the paper. At a cursory study, it will appear that the authors did not assess external validity (Predictive, convergent or discriminant). However, a careful read shows these tests were done. Differentiating between the different forms of validity will enhance the logical progression of the paper. Please refer to:

 2.D.3. Author Response: We have updated the methods section referencing validity more specifically to discuss convergent and discriminant validity. Note we did not assess predictive validity in this cross-sectoinal assessment. Updated information is as follows:

‘As previously noted, content validity was assessed by two external experts and the two Research Assistants (RAs) who led Free-list interviews and FGDs, and face validity was assessed by both the RAs and the enumerator team. Convergent validity was determined by assessing correlations between each subscale score and items from the survey that asked about thematically similar concepts. Specifically, for Subscale 1 (Restrictions), we assessed correlation of scores with traveling alone; Subscale 2 (Management) scores with having resources to change menstruation material; Subscale 3 (Symptoms) scores with infection symptoms, and Subscale 4 (Menstrual Cycle Concerns) scores with experiencing regular menstruation. Additionally, known-groups validation was used to assess discriminant validity, with two sample t-tests being used to verify whether Subscale 5 (Menstruation-Related Bodily Concerns) scores differed by those reporting burning or itching in the vaginal area.’ (lines 486-496)

2.D.4. The authors do indicate that the amount of variation explained by the 19 items is above the 60% threshold. It is not clear what threshold was used for this assessment. The authors should add a citation to back this position. To my knowledge, 70% is the threshold used in most psychometric evaluations. I should indicate that the current variation of 69% is as good as 70%.

 2.D.4. Author Response: We have added the following citation:

Hair, J. F. Jr., Black, W. C., Babin, B. J., & Anderson, R. E. (2014). Multivariate Data Analysis (Seventh Edition ed.). Essex, England: Pearson Education Limited.

Text from citation: “Rule of thumb:Choosing Factor Models and Number of Factors…..Enough factors to meet a specified percentage of variance explained, usually 60% or higher”.

2.D.5. Due to the multidimensionality of the scale items, it is possible that all the subdomains could form a unidimensional scale. i.e. all 19 items could be assessed to be unidimensional or form an overarching measure. Hierarchical confirmatory factor analysis is necessary to determine whether this is the case. Also, if the authors intend to use each of the subscales by themselves, then they need to assess whether each of the subscales is unidimensional in the second data. Further, it is possible that factor 3 may be contributing noise to the entire model. The authors may need to test for this using a bifactor model to be sure this is not the case.

2.D.5. Author Response: We have made edits to the manuscript in both the methods and results sections indicating that we used Hierarchical Confirmatory Factor analysis to confirm that sub-scales form a unidimensional scale:

Added to methods (phase 3, stage 3) :

‘Hierarchical confirmatory factor analysis was used to confirm that sub-scales form a unidimensional scale.’ (lines 479-480)

Added to results:

Hierarchical CFA confirmed that all sub-scales form a unidimensional scale, and that a total score can be used to assess menstrual insecurity (RMSEA = 0.057; CFI = 0.936; TLI = 0.926; See Supplemental Figure 2 for Hierarchical CFA Model Diagram). Additionally, bifactor CFA models with and without factor 3 were used to confirm its psychometric relevance to the overall factor structure, despite having the lowest factor variance in EFA.’ (lines 660-665)

2.D.6.The test of reliability makes one to wonder whether the authors tested for the unidimensionality of the entire 19-item scale. If that was not done, it is difficult to understand why they assessed the reliability of the 19 items together. Emphasis should be laid on the reliability of the sub scales or factors. Again, if this test was not done, the authors cannot use the total score of all 19 items as a menstruation insecurity score. Consequently, the need to do a higher order confirmatory factor analysis and test for the unidimensionality of the scale items.

2.D.6. Author Response: Hierarchical CFA (as described above) has confirmed that assessing full scale reliability is appropriate.

2.D.7. The diagrammer in Mplus produces one of the best figures associated with CFA. The authors should consider providing a diagram output for clarity of the results.

2.D.7. Author Response: We have added a digram of the Hierarchical CFA model to the supplement so that readers can have additional information on that specific odel. We did not create one for the first order CFA as that information is already provided in tables in the manuscript itself.

2.D.8. The authors assert that the mean overall Menstruation Insecurity score was 1.46; the authors should clarify whether this is coming from an average of all 19 items. If so, it needs to be justified as confirmatory factor analysis was not conducted directly on all 19 items but sub-factors.

2.D.8. Author Response: Hierarchical CFA (as described above) has confirmed that assessing full scale mean score is appropriate.

2.D.9. For the construct validity, the authors should consider using a regression model instead of correlations. At minimum, a bivariate regression model. This will also provide the amount of variance explained by each construct in the outcome variable.

 2.D.9. Author Response: Using correlations to assess construct validity is standard practice and we are confident in our approach. As is noted by Jepson (2017) ‘Construct validity is the extent to which an instrument assesses a construct of concern and is associated with measures of other constructs in that domain.’ We have used correlations to identify associations. If we were assessing predictive validity, we would have an outcome of interest and would use regression models.

Jepson, W. E., Wutich, A., Colllins, S. M., Boateng, G. O., & Young, S. L. (2017). Progress in household water insecurity metrics: a cross‐disciplinary approach. Wiley Interdisciplinary Reviews: Water4(3), e1214.

2.D.10. Again, the total score as indicated in Table 4 is not possible if the authors did not test for the unidimensionality of all the 19 items. If no new models are added, the authors may have to revise the discussion to show the value of the menstruation insecurity scale based on the sub-scales.

2.D.10. Author Response: Hierarchical CFA (as described above) has confirmed that assessing full scale mean score is appropriate.

2.D.11. The authors may have to explain in the domain identification how they arrived theoretically at the five domains. Even if it is reported elsewhere, it is important that this is made clear in the paper.

2.D.11. Author Response: We have made amendments to the text.

First, in the methods [Section 2.4.2. Phase 1, Stage 2: Item Identification], we added more information about how we initially identified four themes from the qualitative work that we expected to be factors, and included a supplement as well:

‘All FLIs were analyzed first to identify the full suite of concerns or negative experiences participants believed women in their community to have. We then determined the proportion of participants who reported each concern or experience. Thirty-two unique menstruation concerns were shared by 67 (97%) of the 69 women interviewed. FGDs supported information gathered in FLIs (See Caruso 2017, including supplemental tables, for list and frequencies of concerns)[22]. Next, concerns were analyzed thematically to identify emergent grouping. We identified and sorted concerns into four themes, which included: Management, Restrictions or challenges to normal activities, Social needs and constraints, and Well-Being. Wording of the identified concerns in each theme was then adapted to create draft items. (See Supplement Table 1 for Survey Items by theme) The initial list of items included the full range of concerns noted, however monsoon-specific items were omitted due to irrelevance during the following anticipated data collection period. We hypothesized that these themes would emerge as factors in the final measure.’ (lines 317-340)

Further, we have expanded the results [Section 3.2. Exploratory factor analysis] with further elaboration:

‘Based on the items within, we labeled the five factors: “Restrictions,” “Management”, “Symptoms,” “Menstrual Cycle Concerns,” and “Menstruation-Related Bodily Concerns” (referred to moving forward as ‘Bodily Concerns’ for ease). Factor 1 (Restrictions) includes two items about concerns continuing regular activities during menstruation (factor loadings: 0.866-0.998; 9.4% variance explained). Factor 2 (Management) contains nine items about women’s concerns about their ability to access, wash, and store their chosen materials and care for themselves and their menstrual needs during menstruation (factor loadings: 0.482-0.932; 27.1% variance explained). Factor 3 (Symptoms) contains two items about physical symptoms experienced during menstruation (factor loadings: 0.847-883; 8.3% variance explained). Factor 4 (Menstrual Cycle Concerns) contains three items about women’s concerns about their menstrual cycle or overall health related to their menstruation (factor loadings: 0.800-1.035; 12.7% variance explained). Finally, Factor 5 (Bodily Concerns) includes three items that address the concerns or experiences related to the body when using menstruation materials (factor loadings: 0.580-1.058; 11.6% variance explained). As hypothesized, the ‘Management’ and ‘Restrictions’ factors were retained. Our hypothesized ‘Well-Being’ factor essentially became two factors, Symptoms and Menstrual Cycle Concerns. The Bodily Concerns factor emerged to include items from three of the hypothesized factors.’ (lines 604-619)

2.D.12. While I appreciate the value of a menstruation insecurity scale, menstruation is a biological phenomenon and not a latent construct. To measure it as a latent construct, the concept needs to be used in relation to a qualifier or context. Hence, I will ask that the authors consider using menstruation hygiene insecurity instead of menstruation insecurity. Alternatively, the authors should clarify the conceptualization of menstruation insecurity.

2.D.12. Author Response: We understand this point and have changed the name to ‘Menstrual Insecurity’. We do not want to use the term ‘Menstruation Hygiene Insecurity’ as that is a bit confining and suggests a focus on management.

2.D.13 The authors in the third paragraph of the discussion indicate that “our work shows that menstruation can impact women’s ability to work, an understudied area of menstruation research”. However, this is based on the comparison between known groups, which did not adjust for any control variables. The relationship between the variables could be spurious. Hence, it is best to showcase the significant differences and not draw definite conclusions.

 2.D.13. Author Response: We agree and have softened the language in this specific sentence to now read as follows:

‘We found that those who reported difficulty doing regular work during menstruation had significantly higher overall Menstrual Insecurity and individual sub-scale scores than those reporting they never faced difficulties.’ (lines 831-833)

At the end of the paragraph, we also noted that our analyses were limited to known groups and that further research was warranted.

2.E. Interest to the Readers

2.E.1 This paper has significant relevance to researchers in the area of water, sanitation and hygiene and has a lot of relevance in measuring health inequities for women in resource poor settings. Due to its novelty, it is very likely that academics, public health practitioners, community health workers, and policy makers will find the tool useful in promoting the health of women of all ages, especially, those that have reached menarche. I have no doubt that this paper will attract a wider readership.

 2.E.1. Author Response: We are encouraged by the reviewer’s positive response to this paper. We hope this work encourages further measurement development in the sector.

2.F. Overall Merit

2.F.1 This study makes an important contribution to research in the area of water, sanitation and hygiene, women's health, and produces a unique tool which can used to assess health inequities, challenges of women in relation to menstruation, and the correlates of menstruation insecurity.

2.F.1. Author Response: Again, we thank the reviewer for their time reviewing this paper and appreciate how the comments have strengthened this work.

2.G. English level

2.G.1. The paper is well written and logically coherent in structure. However, I will ask that the authors are consistent in their use of menarche or period. You may choose to stay with a single description for menstruation. Please pay attention to subject verb agreements e.g. Line 107, line 221

2.G.1. Author Response: We believe the language is now more consistent and we have also reviewed for grammar/ verb agreement as suggested.

Round 2

Reviewer 1 Report

Revisions have improved the paper.

There are a few outstanding revisions required before this can be published.

The revisions to the title don't seem to be showing up in the journal system? - I have taken the revisions on the manuscript document as the correct versions.

The revised definition of menstrual insecurity is clearer. Although I have ongoing concerns about how authors can reasonably test the validity of a measure for a concept that still requires more clarity, I accept that they have noted that this needs to be developed further in future work.

Notably in addressing my original concerns about construct clarity authors need to make clear in the paper that as the concept of menstrual insecurity is developed this will have implications for content validity of this measure. Content validity requires that the items on a test "are representative of the domain [construct] the test seeks to measure" - this is difficult (impossible?) to appraise without further clarity on menstrual insecurity. What needs to be included in the scale for it to have content validity and represent menstrual insecurity may evolve with the definition of the construct. This can be noted as appropriate throughout, in limitations - or may be useful to address near line 538 where authors rightfully point out that other rejected items may also be relevant to measurement in menstrual health/hygiene research.

I appreciate that the authors have revised the structure and clarified language about the nature of post-hoc analyses with rejected scale items. I do still find this concerning from a conceptual point of view. 

Given that these are post-hoc analyses and that more a-priori validation was undertaken with different item these should NOT be included in the abstract (line 28-30). Authors should either strongly clarify this line of the abstract along the lines of 'post hoc exploratory analyses found...', or better replace this with the stronger analyses with other items used to test validity across the sub-scales. 

Line 48 needs to be revised to 87 publications, not studies, as in line 65.

Line 93 - isn't the correct spelling "complement" not "compliment" when used in this way?

Line 307 - This should be corrected to "Post-hoc exploration of Menstrual Insecurity Scores" for clarity - or two sub-sections made of the two following paragraphs. 

Line 311-312 -Plenty of women describe experiencing tension or emotional changes during their period, it should be clarified that contextually tension was understood in relation to anxiety about menstruation, not emotional variations due to hormones - dismissing this as only occurring before menstruation doesn't seem sufficient. 

Line 523 - A menstrual symptoms questionnaire already exists and symptoms are also the main component of the Menstrual Distress Questionnaire (which I believe has been investigated across a few contexts, although obviously more culturally appropriate versions for some locations and an update is likely useful)

Stephenson, L. A., Denney, D. R., & Aberger, E. W. (1983). Factor structure of the menstrual symptom questionnaire: Relationship to oral contraceptives, neuroticism and life stress. Behaviour research and therapy21(2), 129-135.

Moos, R. H. (1968). The development of a menstrual distress questionnaire. Psychosomatic medicine30(6), 853-867.

Line 572 - authors address (in revisions and in their response to reviewers) the high proportion of 'never' in the Menstrual Cycle Concerns sub-scale. But this is also true for the 'Management' sub-scale for which many items have >90% 'never' endorsement. This also needs to be addressed before the manuscript is published, is the item 'difficulty' here too low for the population? Do these quantitative results suggest that management concerns raised qualitatively are overblown? The high 'never' reporting in this sub-scale needs to be addressed by the authors and flagged appropriately for readers considering use of the scale.

Author Response

7 May 2020

To Whom It May Concern:

RE: ‘Assessing Women’s Menstruation Concerns and Experiences in Rural India: Development and Validation of a Menstrual Insecurity Measure’ (manuscript: ijerph-774592).

We appreciate the time provided in reviewing our manuscript. We have provided a point-by-point response to all outstanding comments from the reviewer. We have copied refered text sections into the responses below to enable an easier review, and have also made all changes in track changes on the manuscript document for reviewing ease.

We hope you find these revisions acceptable. Please do not hesitate to contact me with any questions.

Sincerely,

Bethany A. Caruso 

Reviewer 1 Overall Comments and Suggestions for Authors

  1. Revisions have improved the paper. There are a few outstanding revisions required before this can be published.

Author Response: We agree that revisions have improved the paper and are grateful for the comments provided to enable this strengthening. We have responded to the additional comments and hope they address all concerns appropriately.

  1. The revisions to the title don't seem to be showing up in the journal system? - I have taken the revisions on the manuscript document as the correct versions.

Author Response: The version on the document is the title we would like to use moving forward.

  1. The revised definition of menstrual insecurity is clearer. Although I have ongoing concerns about how authors can reasonably test the validity of a measure for a concept that still requires more clarity, I accept that they have noted that this needs to be developed further in future work.

Notably in addressing my original concerns about construct clarity authors need to make clear in the paper that as the concept of menstrual insecurity is developed this will have implications for content validity of this measure. Content validity requires that the items on a test "are representative of the domain [construct] the test seeks to measure" - this is difficult (impossible?) to appraise without further clarity on menstrual insecurity. What needs to be included in the scale for it to have content validity and represent menstrual insecurity may evolve with the definition of the construct. This can be noted as appropriate throughout, in limitations - or may be useful to address near line 538 where authors rightfully point out that other rejected items may also be relevant to measurement in menstrual health/hygiene research.

Author Response: The following has been added to the limitations section to address this point:

Finally, as research on menstruation evolves, the definition of Menstrual Security, which is currently broad, may also evolve. Any future development of Menstrual Insecurity as a concept will have implications for the content validity of the current measure. In other words, the current items and sub-scales may not fully represent a future iteration of Menstrual Insecurity. Still, we consider this work and this measure to be a starting point, and accept that the concept may evolve and that the measure may need to evolve as well. Indeed, concepts of food and water insecurity have changed over time, and so too have their measures, and those fields have benefited from those evolutions.’

  1. I appreciate that the authors have revised the structure and clarified language about the nature of post-hoc analyses with rejected scale items. I do still find this concerning from a conceptual point of view. 

Given that these are post-hoc analyses and that more a-priori validation was undertaken with different item these should NOT be included in the abstract (line 28-30). Authors should either strongly clarify this line of the abstract along the lines of 'post hoc exploratory analyses found...', or better replace this with the stronger analyses with other items used to test validity across the sub-scales. 

Author Response: We have clarified this line in the abstract as recommended. The line in the abstract now reads as follows:

‘Post hoc exploratory analysis found that women reporting experiencing tension at menstrual onset or difficulty doing work had significantly higher Menstrual Insecurity scores.’ 

  1. Line 48 needs to be revised to 87 publications, not studies, as in line 65.

Author Response: The line now reads as follows:

‘A 2019 systematic review synthesizing findings from 87 qualitative publications on women’s and girl’s menstruation experiences in LMICs illuminates how these menstruation experiences can have broader impacts.’

  1. Line 93 - isn't the correct spelling "complement" not "compliment" when used in this way?

Author Response: The line now reads as follows:

‘We refer to the complement of concerns and negative menstruation-related experiences women in this population face as ‘Menstrual Insecurity’.

  1. Line 307 - This should be corrected to "Post-hoc exploration of Menstrual Insecurity Scores" for clarity - or two sub-sections made of the two following paragraphs. 

Author Response: The line now reads as follows:

Post-hoc exploration of Menstrual Insecurity scores were highest among those reporting tension at menstrual onset or difficulty doing regular work during menstruation, and suggest the need for further research.’

  1. Line 311-312 -Plenty of women describe experiencing tension or emotional changes during their period, it should be clarified that contextually tension was understood in relation to anxiety about menstruation, not emotional variations due to hormones - dismissing this as only occurring before menstruation doesn't seem sufficient. 

Author Response: We added the following sentence and reference:

‘For clarity, in this context tension is understood to be distress about menstruation, not emotional variations due to hormones, similar to what has been described by Weaver (2017)[58].’

Weaver, L.J. Tension among women in North India: An idiom of distress and a cultural syndrome. Culture, Medicine, and Psychiatry 2017, 41, 35-55.

  1. Line 523 - A menstrual symptoms questionnaire already exists and symptoms are also the main component of the Menstrual Distress Questionnaire (which I believe has been investigated across a few contexts, although obviously more culturally appropriate versions for some locations and an update is likely useful)

Stephenson, L. A., Denney, D. R., & Aberger, E. W. (1983). Factor structure of the menstrual symptom questionnaire: Relationship to oral contraceptives, neuroticism and life stress. Behaviour research and therapy21(2), 129-135.

Moos, R. H. (1968). The development of a menstrual distress questionnaire. Psychosomatic medicine30(6), 853-867.

Author Response: Thus far, the lines that the reviewer has suggested have been a bit off from the current version of the paper. We have been able to use the comment context to discern what the reviewer suggest we fix and where. For this comment, however, it is not clear what the reviewr is suggesting. Is the suggestion that we cite this work? These two papers are quite dated, as the reviewer notes, and the items (like use of prescription drugs for pain) are not appropriate for the context. We have added the following and hope it addresses the concern:

‘Further, we think it also would be useful to test a culturally appropriate ‘Menstruation Symptoms Index’ to accompany this sub-scale. The index could be used to determine which, of a range of symptoms, women experience, and how intense the symptoms are. Some older measures do exist, and these would need testing to assure contextual appropritateness [63,64].’

  1. Moos, R.H. The development of a menstrual distress questionnaire. Psychosomatic medicine 1968, 30, 853-867.
  2. Stephenson, L.A.; Denney, D.R.; Aberger, E.W. Factor structure of the menstrual symptom questionnaire: Relationship to oral contraceptives, neuroticism and life stress. Behaviour research and therapy 1983, 21, 129-135.

  1. Line 572 - authors address (in revisions and in their response to reviewers) the high proportion of 'never' in the Menstrual Cycle Concerns sub-scale. But this is also true for the 'Management' sub-scale for which many items have >90% 'never' endorsement. This also needs to be addressed before the manuscript is published, is the item 'difficulty' here too low for the population? Do these quantitative results suggest that management concerns raised qualitatively are overblown? The high 'never' reporting in this sub-scale needs to be addressed by the authors and flagged appropriately for readers considering use of the scale.

 Author Response: We have added the following to the discussion:

Similarly, for four items in the management factor, 90% or more of respondents indicated that they ‘never’ had the concern. Each of the four items queried about cloth or pads, specifically difficulty in accessing materials needed, or difficulty finding a place to dispose, change, or store materials. Given that a high proportion (70%) of the sample reported using resubale materials, it is not surprising that disposal was never an issue for 90% of respondents. For the other items, we simply ask about difficulty in accessing materials needed or in finding places for changing and storing in general. These items may have been too broad, too general. In the qualitative work, women discussed not being able to get materials they preferred, or changing or storing materials in places the felt were not ideal. In future iterations of the scale, we would recommend refining these items to ask about perceived suitability of the materials accessed or places used for management. Women are resilient and will find a means to address their needs, but they may not do so in a manner that aligns with their preferences.’